# GraphFLEx: Structure Learning Framework for Large Expanding Graphs

## Abstract

Graph structure learning is a core problem in graph-based machine learning, essential for uncovering latent relationships and ensuring model interpretability. However, most existing approaches are ill-suited for large-scale and dynamically evolving graphs, as they often require complete re-learning of the structure upon the arrival of new nodes and incur substantial computational and memory costs. In this work, we propose GraphFLEx—a unified and scalable framework for Graph Structure Learning in Large and Expanding Graphs. GraphFLEx mitigates the scalability bottlenecks by restricting edge formation to structurally relevant subsets of nodes identified through a combination of clustering and coarsening techniques. This dramatically reduces the search space and enables efficient, incremental graph updates. The framework supports 48 flexible configurations by integrating diverse choices of learning paradigms, coarsening strategies, and clustering methods, making it adaptable to a wide range of graph settings and learning objectives. Extensive experiments across 26 diverse datasets and Graph Neural Network architectures demonstrate that GraphFLEx achieves state-of-the-art performance with significantly improved scalability. Our implementation is publicly available here.

## 1 Introduction

Graph representations capture relationships between entities, vital across diverse fields like biology, finance, sociology, engineering, and operations research [1–4]. While some relationships, such as social connections or sensor networks, are directly observable, many, including gene regulatory networks, scene graph generation [5], brain networks, [6] and drug interactions, require inference [7]. Even when available, graph data often contains noise, requiring denoising and recalibration. In such cases, inferring the correct graph structure becomes more crucial than the specific graph model or downstream algorithm.

*Graph Structure Learning (GSL)* offers a solution, enabling the construction and refinement of graph topologies. GSL has been widely studied in both supervised and unsupervised contexts [8, 9]. In supervised GSL (s-SGL), the adjacency matrix and Graph Neural Networks (GNNs) are jointly optimized for a downstream task, such as node classification. Notable examples of s-GSL include $NodeFormer$ [10], $Pro-GNN$ [11], $WSGNN$ [12], and $SLAPS$ [13]. Unsupervised GSL (u-SGL), on the other hand, focuses solely on learning the underlying graph structure, typically through adjacency or Laplacian matrices. Methods in this category include approximate nearest neighbours ($A-NN$) [14, 15], k-nearest neighbours ($k-NN$) [16, 17], covariance estimation ($emp.Cov.$) [18], graphical lasso ($GLasso$) [19], $SUBLIME$ [8], and signal processing techniques like $l2$-model, $log$-model and $large$-model [20, 21].

Supervised structure learning (s-SGL) methods have demonstrated effectiveness in specific tasks; however, their reliance on labeled data and optimization for downstream objectives—particularly node classification—significantly constrains their generalizability to settings where annotations are scarce or unavailable [8]. Unsupervised structure learning (u-SGL) methods, which constitute the focus

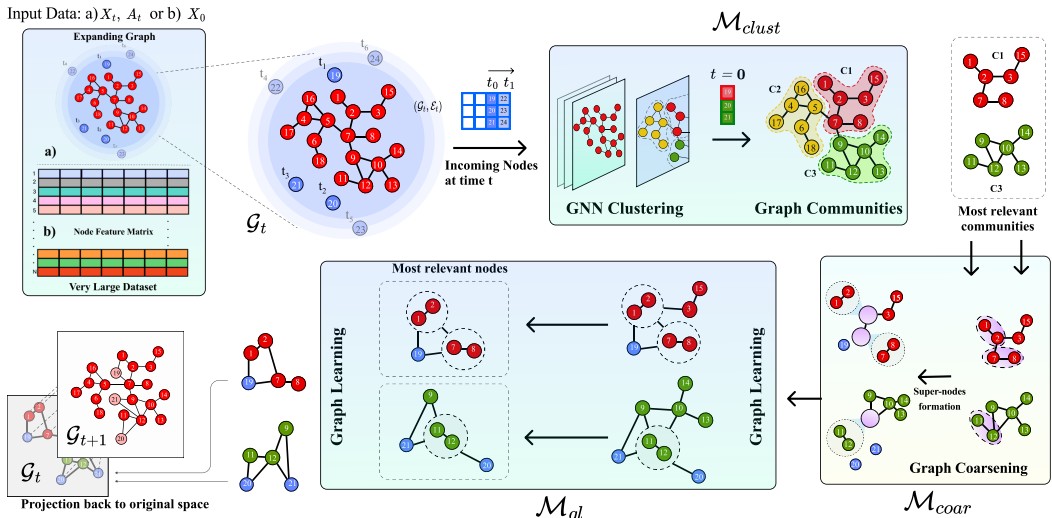

Figure 2: This figure illustrates the general pipeline of GraphFLEx, designed to efficiently handle both a) large datasets with missing structure and b) expanding graphs. Both scenarios can be modeled as expanding graphs (details in Section 3.1). GraphFLEx processes a graph ($\mathcal{G}_t$) and incoming nodes ($\mathcal{E}_{t+1}$) at time $t$, newly arriving nodes are shown with different timestamps and shades of blue to indicate their arrival time. Our framework comprises of three main components: i) **Clustering**, which infers $\mathcal{E}_{t+1}$ nodes to existing communities using a pre-trained model $\mathcal{M}_{\text{clust}}(\mathcal{G}_0)$ into smaller, more manageable communities; ii) Since these communities may still be large, a **Coarsening**, module is applied to further reduce their size while preserving essential structural information; and iii) Finally, a **Learning** module, where the structure associated with $\mathcal{E}_{t+1}$ nodes are learned using the coarsened graph, followed by projecting this structure onto the $\mathcal{G}_t$ graph to create graph $\mathcal{G}_{t+1}$.

of this work, offer broader applicability. Nevertheless, both s-SGL and u-SGL approaches exhibit critical limitations in their ability to scale to large graphs or adapt efficiently to expanding datasets.

To address these challenges, we introduce **GraphFLEx**, a unified and scalable framework for *Graph Structure Learning in Large and Expanding Graphs*. GraphFLEx is built upon the coordinated integration of three foundational paradigms in graph processing: *graph clustering*, *graph coarsening*, and *structure learning*. While each of these methodologies has been studied extensively in isolation, their joint application within a single framework has remained largely unexplored. The novelty of GraphFLEx lies not merely in combining these components, but in the principled manner in which they are algorithmically aligned to reinforce one another—clustering serves to localize the search space, coarsening reduces structural redundancy while preserving global properties, and structure learning operates efficiently within this refined context. This integration enables GraphFLEx to scale effectively to large datasets and accommodate dynamic

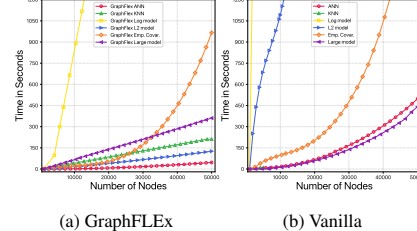

|     (a) GraphFLEx     |     (b) Vanilla     |

Figure 1: High computational time required to learn graph structures using existing methods, whereas GraphFLEx effectively controls computational growth, achieving near-linear scalability. Notably, Vanilla KNN failed to construct graph structures for more than 10K nodes due to memory limitations.

graphs through *incremental updates*, eliminating the need for expensive re-training. Additionally, the framework supports **48 modular configurations**, enabling broad adaptability across datasets, learning objectives, and deployment constraints. Crucially, we establish *theoretical guarantees* on edge recovery fidelity and computational complexity, offering rigorous foundations for the framework's efficiency and reliability. As illustrated in Figure 2, GraphFLEx significantly reduces the candidate edge space by operating on structurally relevant node subsets. Empirical evaluations, summarized in Figure 1, demonstrate that GraphFLEx substantially outperforms existing baselines in both runtime and scalability.

**Key contributions of this work include:**

- GraphFLEx unifies *multiple structure learning strategies* within a single flexible framework.
- GraphFLEx demonstrates effectiveness in *handling growing graphs*.
- GraphFLEx enhances the *scalability* of graph structure learning on large-scale graphs.
- GraphFLEx serves as a *comprehensive framework* applicable individually for clustering, coarsening, and learning tasks.

- We provide both *empirical and theoretical results*, demonstrating the effectiveness of GraphFLEx across a range of datasets.

## 2 Problem Formulation and Background

A graph $\mathcal{G}$ is represented using $\mathcal{G}(V, A, X)$ where $V = \{v_1, v_2...v_N\}$ is the set of $N$ nodes, each node $v_i$ has a $d-$dimensional feature vector $x_i$ in $X \in \mathbb{R}^{N \times d}$ and $A \in \mathbb{R}^{N \times N}$ is adjacency matrix representing connection between $i^{th}$ and $j^{th}$ nodes when entry $A_{ij} > 0$. An expanding graph $\mathcal{E}_\mathcal{G}$ can be considered a variant of graph $\mathcal{G}$ where nodes $v$ now have an associated timestamp $\tau_v$. We can represent a expanding graph as a sequence of graphs, i.e., $\mathcal{E}_\mathcal{G} = \{\mathcal{G}_0, \mathcal{G}_1, ...\mathcal{G}_T\}$ where $\{\mathcal{G}_0 \subseteq \mathcal{G}_1.... \subseteq \mathcal{G}_T\}$ at $\tau \in \{0, ...T\}$ timestamps. New nodes arriving at different timestamps are seamlessly integrating into initial graph $\mathcal{G}_0$.

***Problem statement.*** Given a partially known or missing graph structure, our goal is to incrementally learn the whole graph, i.e., learn adjacency or laplacian matrix. Specifically, we consider two unsupervised GSL tasks:

**Goal 1.** *Large Datasets with Missing Graph Structure: In this setting, the graph structure is entirely unavailable, and existing methods are computationally infeasible for learning the whole graph in a single step. To address this issue, we first randomly partition the dataset into exclusive subsets. We then learn the initial graph $\mathcal{G}_0(V_0, X_0)$ over a small subset of nodes and incrementally expand it by integrating additional partitions, ultimately reconstructing the full graph $\mathcal{G}_T$.*

**Goal 2.** *Partially Available Graph: In this case, we only have access to the graph $\mathcal{G}_t$ at timestamp $t$, with new nodes arriving over time. The goal is to update the graph incrementally to obtain $\mathcal{G}_T$, without re-learning it from scratch at each timestamp.*

GraphFlex addresses these challenges with a unified framework, outlined in Section 3. Before delving into the framework, we review some key concepts.

### 2.1 Graph Reduction

Graph reduction encompasses sparsification, clustering, coarsening, and condensation [22]. Graph-Flex employs clustering and coarsening to refine the set of relevant nodes for potential connections.
**Graph Clustering.** Graphs often exhibit global heterogeneity with localized homogeneity, making them well-suited for clustering [23]. Clusters capture higher-order structures, aiding graph learning. Methods like DMoN [24] use GNNs for soft cluster assignments, while Spectral Clustering (SC) [25] and K-means [16, 26] efficiently detect communities. DiffPool [27, 28] applies SC for pooling in GNNs.
**Graph Coarsening.** Graph Coarsening (GC) reduces a graph $\mathcal{G}(V, E, X)$ with $N$ nodes and features $X \in \mathbb{R}^{N \times d}$ into a smaller graph $\mathcal{G}_c(\widetilde{V}, \widetilde{E}, \widetilde{X})$ with $n \ll N$ nodes and $\widetilde{X} \in \mathbb{R}^{n \times d}$. This is achieved via learning a coarsening matrix $\mathcal{P} \in \mathbb{R}^{n \times N}$, mapping similar nodes in $\mathcal{G}$ to super-nodes in $\mathcal{G}_c$, ensuring $\widetilde{X} = \mathcal{P}X$ while preserving key properties [29–32].

### 2.2 Unsupervised Graph Structure Learning

Unsupervised graph learning spans from simple k-NN weighting [17, 33] to advanced statistical and graph signal processing (GSP) techniques. Statistical methods, also known as probabilistic graphical models, assume an underlying graph $\mathcal{G}$ governs the joint distribution of data $X \in \mathbb{R}^{N \times d}$ [19, 34, 35]. Some approaches [36] prune elements in the inverse sample covariance matrix $\widehat{\Sigma} = \frac{1}{d-1} X X^T$ and sparse inverse covariance estimators, such as Graphical Lasso (GLasso) [19]: $\text{maximize}_\Theta \log \det \Theta - \text{tr}(\widehat{\Sigma}\Theta) - \rho\|\Theta\|_1$, where $\Theta$ is the inverse covariance matrix. However, these methods struggle with small sample sizes. Graph Signal Processing (GSP) techniques analyze signals on known graphs, ensuring properties like smoothness and sparsity. Signal smoothness on a graph $\mathcal{G}$ is quantified by the Laplacian quadratic form: $Q(\mathbf{L}) = \mathbf{x}^T \mathbf{L} \mathbf{x} = \frac{1}{2} \sum_{i,j} w_{ij}(\mathbf{x}(i) - \mathbf{x}(j))^2$. For a set of vectors $X$, smoothness is measured using the Dirichlet energy [37]: $\text{tr}(X^T L X)$. State-of-the-art methods [20, 21, 38] optimize Dirichlet energy while enforcing sparsity or specific structural constraints. Table 7 in Appendix D compares various graph learning methods based on their formulations and time complexities. More recently, SUBLIME [8] learns graph structure in an unsupervised manner by leveraging self-supervised contrastive learning to align a learnable graph with a dynamically refined anchor graph derived from the data itself.

*Remark* 1. Graph Structure Learning (GSL) differs significantly from Continual Learning (CL) [39–41] and Dynamic Graph Learning (DGL) [42–44], as discussed in Appendix C.

# 3 GraphFLEx

In this section, we introduce GraphFLEx, which has three main modules:

- **Graph Clustering.** Identifies communities and extracts higher-order structural information,
- **Graph Coarsening.** Is used to coarsen down the desired community, if the community itself is large,
- **Graph Learning.** Learns the graph's structure using a limited subset of nodes from the clustering and coarsening modules, *enabling scalability*.

For pseudocode, see Algorithm 1 in Appendix G.

## 3.1 Incremental Graph Learning for Large Datasets

Real-world graph data is continuously expanding. For instance, e-commerce networks accumulate new clicks and purchases daily [45], while academic networks grow with new researchers and publications [46]. To manage such growth, we propose incrementally learning the graph structure over smaller segments.

Given a large dataset $\mathcal{L}(V_{\mathcal{L}}, X_{\mathcal{L}})$, where $V_{\mathcal{L}}$ is the node set and $X_{\mathcal{L}}$ represents node features, we define an *expanding dataset* setting $\mathcal{L}_{\mathcal{E}} = \{\mathcal{E}_{\tau=0}^{T}\}$. Initially, $\mathcal{L}$ is split into: (i) a *static dataset* $\mathcal{E}_0(V_0, X_0)$ and (ii) an *expanding dataset* $\mathcal{E} = \{\mathcal{E}_\tau(V_\tau, X_\tau)\}_{\tau=1}^{T}$. Both *Goal 1* (large datasets with missing graph structure) and *Goal 2* (partially available graphs with incremental updates), discussed in Section 2, share the common objective of incrementally learning and updating the graph structure as new data arrives. GraphFLEx handles these by decomposing the problem into two key components:

- **Initial Graph** $\mathcal{G}_0(V_0, A_0, X_0)$**:** For *Goal 1*, where the graph structure is entirely missing, $\mathcal{E}_0(V_0, X_0)$ is used to construct $\mathcal{G}_0$ from scratch using structure learning methods (see Section 2.2). For *Goal 2*, the initial graph $\mathcal{G}_0(V_0, A_0, X_0)$ is already available and serves as the starting point for incremental updates.
- **Expanding Dataset** $\mathcal{E} = \{\mathcal{E}_\tau(V_\tau, X_\tau)\}_{\tau=1}^{T}$**:** In both cases, $\mathcal{E}$ consists of incoming nodes and features arriving over $T$ timestamps. These nodes are progressively integrated into the existing graph, enabling continuous adaptation and growth.

The partition is controlled by a parameter $r$, which determines the proportion of static nodes: $r = \frac{\|V_0\|}{\|V_{\mathcal{L}}\|}$. For example, $r = 0.2$ implies that 20% of $V_{\mathcal{L}}$ is treated as static, while the remaining 80% arrives incrementally over $T$ timestamps. In our experiments, we set $r = 0.5$ and $T = 25$.

*Remark* 2. We can learn $\mathcal{G}_\tau(V_\tau, A_\tau, X_\tau)$ by aggregating $\mathcal{E}_\tau$ nodes in $\mathcal{G}_{\tau-1}$ graph. Our goal is to learn $\mathcal{G}_T(V_T, A_T, X_T)$ after $T^{th}$-timestamp.

## 3.2 Detecting Communities

From the static graph $\mathcal{G}_0$, our goal is to learn higher-order structural information, identifying potential communities to which incoming nodes ($V \in V\tau$) may belong. We train the community detection/clustering model $\mathcal{M}_{\text{clust}}$ once using $\mathcal{G}_0$, allowing subsequent inference of clusters for all incoming nodes. While our framework supports spectral and k-means clustering, our primary focus has been on Graph Neural Network (GNN)-based clustering methods. Specifically, we use DMoN [24, 47, 48], which maximizes spectral modularity. Modularity [49] measures the divergence between intra-cluster edges and the expected number. These methods use a GNN layer to compute the partition matrix $C = \text{softmax}(\text{MLP}(\widetilde{X}, \theta_{\text{MLP}})) \in \mathbb{R}^{N \times K}$, where $K$ is the number of clusters and $\widetilde{X}$ is the updated feature embedding generated by one or more message-passing layers. To optimize the $C$ matrix, we minimize the loss function $\Delta(C; A) = -\frac{1}{2m}\text{Tr}(C^T B C) + \frac{\sqrt{k}}{n}|\Sigma_i C_i^T|_F - 1$, which combines spectral modularity maximization with regularization to prevent trivial solutions, where $B$ is the modularity matrix [24]. Our static graph $\mathcal{G}_0$ and incoming nodes $\mathcal{E}$ follow Assumption 1.

**Assumption 1.** *Based on the well-established homophily principle, which forms the basis of most graph coarsening and learning methods. We assume that the generated graphs adhere to the Degree-Corrected Stochastic Block Model (DC-SBM) [50], where intra-class (or intra-community) links are more likely than inter-class links. DC-SBM, an extension of SBM that accounts for degree heterogeneity, making it a more flexible and realistic choice for real-world networks.*

For more details on DC-SBM, see Appendix A.

**Lemma 1.** $\mathcal{M}_{clust}$ *Consistency. We adopt the theoretical framework of [50] for a DC-SBM with $N$ nodes and $k$ classes. The edge probability matrix is parameterized as $P_N = \rho_N P$, where $P \in \mathbb{R}^{k \times k}$ is a symmetric matrix containing the between/within community edge probabilities and it is independent of $N$, $\rho_N = \lambda_N/N$, and $\lambda_N$ is the average degree of the network. Let*

$\hat{y}_N = [\hat{y}_1, \hat{y}_2, \ldots, \hat{y}_N]$ *denote the predicted class labels, and let* $\hat{C}_N$ *be the corresponding* $N \times k$ *one-hot matrix. Let the true class label matrix is* $C_N$, *and* $\mu$ *is any* $k \times k$ *permutation matrix. Under the adjacency matrix* $A^{(N)}$, *the global maximum of the objective* $\Delta(\cdot; A^{(N)})$ *is denoted as* $\hat{C}_N^*$. *The consistency of class predictions is defined as:*

*1. **Strong Consistency.***

$$P_N \left[ \min_\mu \|\hat{C}_N^* \mu - C_N\|_F^2 = 0 \right] \to 1 \quad as\ N \to \infty,$$

*2. **Weak Consistency.***

$$\forall \varepsilon > 0, P_N \left[ \min_\mu \frac{1}{N} \|\hat{C}_N^* \mu - C_N\|_F^2 < \varepsilon \right] \to 1\ as\ N \to \infty.$$

*where* $\|\cdot\|_F$ *is the Frobenius norm. Under the conditions of Theorem 3.1 from [50]:*

- *The* $\mathcal{M}_{clust}$ *objective is strongly consistent if* $\lambda_N / \log(N) \to \infty$, *and*
- *It is weakly consistent when* $\lambda_N \to \infty$.

*Remark* 3. **Structure Learning within Communities.** In $GraphFLEx$, we focus on learning the structure within each community rather than the structure of the entire dataset at once. Strong consistency ensures perfect community recovery, meaning no inter-community edges exist representing the ideal case. Weak consistency, however, allows for a small fraction ($\epsilon$) of inter-community edges, where $\epsilon$ is controlled by $\rho_n$ in $P_n = \rho_n P$, influencing graph sparsity.

By Lemma 1 and Assumption 1, stronger consistency leads to more precise structure learning, whereas weaker consistency permits a limited number of inter-community edges.

### 3.3 Learning Graph Structure on a Coarse Graph

After training $\mathcal{M}_{clust}$, we identify communities for incoming nodes, starting with $\tau = 1$. Once assigned, we determine significant communities those with at least one incoming node and learn their connections to the respective community subgraphs. For large datasets, substantial community sizes may again introduce scalability issues. To mitigate this, we first coarsen the large community graph into a smaller graph and use it to identify potential connections for incoming nodes. This process constitutes the second module of GraphFLEx, denoted as $\mathcal{M}_{coar}$, which employs LSH-based hashing for graph coarsening. The supernode index for $i^{th}$ node is given as:

$$\mathcal{H}_i = maxOccurance \left\{ \left\lfloor \frac{1}{r} \cdot (\mathcal{W} \cdot X_i + b) \right\rfloor \right\} \tag{1}$$

where $r$ (bin width) controls the coarsened graph size, $\mathcal{W}$ represents random projection matrix, $X$ is the feature matrix, and $b$ is the bias term. For further details, refer to UGC [32]. After coarsening the $i^{th}$ community ($C_i$), $\mathcal{M}_{coar}(C_i) = \{\mathcal{P}_i, S_i\}$ yields a partition matrix $\mathcal{P}_i \in \mathbb{R}^{\|S_i\| \times \|C_i\|}$ and a set of coarsened supernodes ($S_i$), as discussed in Section 2.

To identify potential connections for incoming nodes, we define their neighborhood as follows:

**Definition 1.** *The neighborhood of a set of nodes* $\mathcal{E}_i$ *is defined as the union of the top* $k$ *most similar nodes in* $C_i$ *for each node* $v \in \mathcal{E}_i$, *where similarity is measured by the distance function* $d(v, u)$. *A node* $u \in C_i$ *is considered part of the neighborhood if its distance* $d(v, u)$ *is among the* $k$ *smallest distances for all* $u' \in C_i$.

$$\mathcal{N}_k(\mathcal{E}_i) = \bigcup_{v \in \mathcal{E}_i} \{u \in C_i \mid d(v, u) \leq top\text{-}k[d(v, u') : u' \in C_i]\}$$

**Goal 3.** *The neighborhood of incoming nodes* $\mathcal{N}_k(\mathcal{E}_i)$ *represents the ideal set of nodes where the incoming nodes* $\mathcal{E}_i$ *are likely to establish connections when the entire community is provided to a structure learning framework.. A robust coarsening framework must reduce the number of nodes within each community* $C_i$ *while ensuring that the neighborhood of the incoming nodes is preserved.*

### 3.4 Graph Learning only with Potential Nodes

As we now have a smaller representation of the community, we can employ any graph learning algorithms discussed in Section 2.2 to learn a graph between coarsened supernodes $S_i$ and incoming nodes ($V_\tau^i \in V_\tau$). This is the third module of GraphFLEx, i.e., graph learning; we denote it as $\mathcal{M}_{gl}$. The number of supernodes in $S_i$ is much smaller compared to the original size of the community, i.e., $\|S_i\| \ll \|C_i\|$; scalability is not an issue now. We learn a small graph first using $\mathcal{M}_{gl}(S_i, X_\tau^i) = \widetilde{\mathcal{G}}_\tau^i(V_\tau^c, A_\tau^c)$ where $X_\tau^i$ represents features of new nodes belonging to $i^{th}$

Table 1: Time complexity analysis of GraphFLEx. Here, $N$ is the number of nodes in the graph, $k$ is the number of nodes in the static subgraph used for clustering ($k \ll N$), and $c$ represents the number of detected communities. $k_\tau$ denotes the number of nodes at timestamp $\tau$. Finally, $\alpha = \|S_\tau^i\| + \|\mathcal{E}_\tau^i\|$ is the sum of coarsened and incoming nodes in the relevant community at $\tau$ timestamp.

| | $\mathcal{M}_{clust}$ | $\mathcal{M}_{coar}$ | $\mathcal{M}_{gl}$ | GraphFLEx |
|---|---|---|---|---|
| **Best (kNN-UGC-ANN)** | $\mathcal{O}(k^2)$ | $\mathcal{O}\left(\frac{k_\tau}{c}\right)$ | $\mathcal{O}(\alpha \log \alpha)$ | $\mathcal{O}(k^2 + \frac{k_\tau}{c} + \alpha \log \alpha)$ |
| **Worst (SC-FGC-GLasso)** | $\mathcal{O}(k^3)$ | $\mathcal{O}\left(\left(\frac{k_\tau}{c}\right)^2 \|S_\tau^i\|\right)$ | $\mathcal{O}(\alpha^3)$ | $\mathcal{O}(k^3 + \left(\frac{k_\tau}{c}\right)^2 \|S_\tau^i\| + \alpha^3)$ |

community at time $\tau$, $\widetilde{\mathcal{G}}_\tau^i(V_\tau^c, A^c)$ representing the graph between supernodes and incoming nodes. Utilizing the partition matrix $\mathcal{P}_i$ obtained from $\mathcal{M}_{coar}$, we can precisely determine the set of nodes associated with each supernode. For every new node $V \in V_\tau^i$, we identify the connected supernodes and subsequently select nodes within those supernodes. This subset of nodes is denoted by $\omega_{V_\tau^i}$, the sub-graph associated with $\omega_{V_\tau^i}$ represented by $\mathcal{G}_{\tau-1}^i(\omega_{V_\tau^i})$ then undergoes an additional round of graph learning $\mathcal{M}_{gl}(\mathcal{G}_{\tau-1}^i(\omega_{V_\tau^i}), X_\tau^i)$, ultimately providing a clear and accurate connection of new nodes $V_\tau^i$ with nodes of $\mathcal{G}_{\tau-1}$, ultimately updating it to $\mathcal{G}_\tau$. This multi-step approach, characterized by coarsening, learning on coarsened graphs, and translation to the original graph, ensures scalability.

**Theorem 1.** *Neighborhood Preservation. Let $\mathcal{N}_k(\mathcal{E}_i)$ denote the neighborhood of incoming nodes $\mathcal{E}_i$ for the $i^{th}$ community. With partition matrix $\mathcal{P}_i$ and $\mathcal{M}_{gl}(S_i, X_\tau^i) = \mathcal{G}_\tau^c(V_\tau^c, A_\tau^c)$ we identify the supernodes connected to incoming nodes $\mathcal{E}_i$ and subsequently select nodes within those supernodes; this subset of nodes is denoted by $\omega_{V_\tau^i}$. Formally,*

$$\omega_{V_\tau^i} = \bigcup_{v \in \mathcal{E}_i} \left\{ \bigcup_{s \in S_i} \{\pi^{-1}(s) | A_\tau^c(v, s) \neq 0\} \right\}$$

*Then, with probability $\Pi_{\{c \in \phi\}} p(c)$, it holds that $\mathcal{N}_k(\mathcal{E}_i) \subseteq \omega_{V_\tau^i}$ where*

$$p(c) \leq 1 - \frac{2}{\sqrt{2\pi}} \frac{c}{r} \left[ 1 - e^{-r^2/(2c^2)} \right],$$

*and $\phi$ is a set containing all pairwise distance values ($c = \|v - u\|$) between every node $v \in \mathcal{E}_i$ and the nodes $u \in \omega_{V_\tau^i}$. Here, $\pi^{-1}(s)$ denotes the set of nodes mapped to supernode $s$, $r$ is the bin-width hyperparameter of $\mathcal{M}_{coar}$.*

*Proof.* The proof is deferred in Appendix B. $\qquad\square$

*Remark* 4. Theorem 1 establishes that, with a constant probability of success, the neighborhood of incoming nodes $\mathcal{N}_k(\mathcal{E}_i)$ can be effectively recovered using the GraphFLEx multistep approach, which involves coarsening and learning on the coarsened graph, i.e., $\mathcal{N}_k(\mathcal{E}_i) \subseteq \omega_{V_\tau^i}$. The set $\omega_{V_\tau^i}$, estimated by GraphFLEx, identifies potential candidates where incoming nodes are likely to connect. The probability of failure can be reduced by regulating the average degree of connectivity in $\mathcal{M}_{gl}(S_i, X_\tau^i) = \mathcal{G}_\tau^c(V_\tau^c, A_\tau^c)$. While a fully connected network $\mathcal{G}_\tau^c$ ensures all nodes in the community are candidates, it significantly increases computational costs for large communities.

### 3.5 GraphFLEx: Multiple SGL Frameworks

Each module in Figure 3 controls a distinct aspect of the graph learning process: clustering influences community detection, coarsening reduces graph complexity via supernodes, and the learning module governs structural inference. Altering any of these modules results in a new graph learning method. Currently, we support 48 different graph learning configurations, and this number scales exponentially with the addition of new methods to any module. The number of possible frameworks is given by $\alpha \times \beta \times \gamma$, where $\alpha$, $\beta$, and $\gamma$ represent the number of clustering, coarsening, and learning methods, respectively.

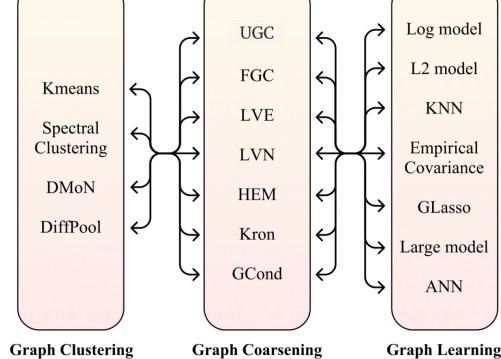

Figure 3: The versatility of GraphFlex in supporting multiple GSL methods.

### 3.6 Run Time Analysis

GraphFLEx computational time is always bounded by existing approaches, as it operates on a significantly reduced set of nodes. We evaluate the run-time complexity of GraphFLEx in two

scenarios: (a) the worst-case scenario, where computationally intensive clustering and coarsening modules are selected, providing an upper bound on time complexity, and (b) the best-case scenario, where the most efficient modules are chosen. Table 1 presents a summary of this analysis for both cases. Due to space limitations, a more comprehensive analysis is provided in Appendix E.

## 4 Experiments

**Tasks and Datasets.** To validate GraphFLEx's utility, we evaluate it across four key dimensions: (i) computational efficiency, (ii) scalability to large graphs, (iii) quality of learned structures, and (iv) adaptability to dynamically growing graphs. To validate the characteristics of GraphFLEx, we conduct extensive experiments on 26 different datasets, including (a) datasets that already have a complete graph structure (allowing comparison between the learned and the original structure), (b) datasets with missing graph structures, (c) synthetic datasets, (d) small datasets for visualizing the graph structure, and (e) large datasets, including datasets with even $2.4M$ nodes. More details about datasets and *system specifications* are presented in Table 8 in Appendix F.

Table 2: Computational time(in seconds) for learning graph structures using GraphFLEx (GFlex) with existing methods (Vanilla referred to as Van.). The experimental setup involves treating 50% of the data as static, while the remaining 50% of nodes are treated as incoming nodes arriving in 25 different timestamps. The best times are highlighted by color Green. OOM and OOT denote out-of-memory and out-of-time, respectively.

| Data | ANN Van. | ANN GFlex | KNN Van. | KNN GFlex | log-model Van. | log-model GFlex | l2-model Van. | l2-model GFlex | emp-Covar. Van. | emp-Covar. GFlex | large-model Van. | large-model GFlex | Sublime Van. | Sublime GFlex |
|------|------|------|------|------|------|------|------|------|------|------|------|------|------|------|
| Cora | 335 | 100 | 8.4 | 36.1 | 869 | 81.6 | 424 | 55 | 8.6 | 30 | 2115 | 18.4 | 7187 | 493 |
| Citeseer | 1535 | 454 | 21.9 | 75 | 1113 | 64.5 | 977 | 54.0 | 14.7 | 59.2 | 8319 | 43.9 | 8750 | 670 |
| DBLP | 2731 | 988 | OOM | 270 | 77000 | 919 | OOT | 1470 | 359 | 343 | OOT | 299 | OOM | 831 |
| CS | 22000 | 12000 | OOM | 789 | OOT | 838 | 32000 | 809 | 813 | 718 | OOT | 1469 | OOM | 1049 |
| PubMed | 770 | 227 | OOM | 164 | OOT | 176 | OOT | 165 | 488 | 299 | OOT | 262 | OOM | 914 |
| Phy. | 61000 | 21000 | OOM | 903 | OOT | 959 | OOT | 908 | 2152 | 1182 | OOT | 2414 | OOM | 2731 |
| Syn 3 | 95 | 37 | OOM | 30 | 58000 | 346 | 859 | 53 | 88 | 59 | 5416 | 42 | 6893 | 780 |
| Syn 4 | 482 | 71 | OOM | 73 | OOT | 555 | OOT | 145 | 2072 | 1043 | OOT | 392 | OOM | 1896 |

Table 3: Node classification accuracies on different GNN models using GraphFLEx (GFlex) with existing Vanilla (Van.) methods. The experimental setup involves treating 70% of the data as static, while the remaining 30% of nodes are treated as new nodes coming in 25 different timestamps. The best and the second-best accuracies in each row are highlighted by dark and lighter shades of Green, respectively. GraphFLEx's structure beats all of the vanilla structures for every dataset. OOM and OOT denotes out-of-memory and out-of-time respectively.

| Data | Model | ANN Van. | ANN GFlex | KNN Van. | KNN GFlex | log-model Van. | log-model GFlex | l2-model Van. | l2-model GFlex | COVAR Van. | COVAR GFlex | large-model Van. | large-model GFlex | Sublime Van. | Sublime GFlex | Base Struct. |
|------|-------|------|------|------|------|------|------|------|------|------|------|------|------|------|------|------|
| DBLP | GAT | 34.23 | 67.37 | OOM | 69.83 | OOT | 69.83 | OOT | 68.98 | 50.48 | 68.56 | OOT | 66.38 | OOM | 68.32 | 70.84 |
| | SAGE | 34.23 | 69.58 | OOM | 70.28 | OOT | 70.28 | OOT | 70.68 | 51.47 | 70.51 | OOT | 69.32 | OOM | 70.28 | 72.57 |
| | GCN | 34.12 | 69.41 | OOM | 73.39 | OOT | 73.39 | OOT | 73.05 | 51.50 | 71.75 | OOT | 68.55 | OOM | 69.06 | 74.43 |
| | GIN | 34.01 | 69.69 | OOM | 68.19 | OOT | 68.19 | OOT | 73.08 | 52.77 | 72.03 | OOT | 71.18 | OOM | 71.87 | 73.92 |
| CS | GAT | 12.47 | 60.89 | OOM | 61.09 | OOT | 60.95 | 18.64 | 61.06 | 58.96 | 88.06 | OOT | 86.22 | OOM | 64.21 | 60.75 |
| | SAGE | 12.70 | 78.81 | OOM | 79.43 | OOT | 79.06 | 19.24 | 78.94 | 56.97 | 93.30 | OOT | 92.79 | OOM | 78.94 | 80.33 |
| | GCN | 12.59 | 63.81 | OOM | 67.94 | OOT | 69.33 | 19.21 | 66.01 | 58.35 | 91.07 | OOT | 84.85 | OOM | 68.92 | 67.43 |
| | GIN | 13.07 | 77.62 | OOM | 78.41 | OOT | 78.55 | 19.24 | 77.61 | 58.26 | 92.07 | OOT | 86.03 | OOM | 77.61 | 55.65 |
| Pub. | GAT | 49.49 | 83.71 | OOM | 84.60 | OOT | 84.60 | OOT | 84.04 | 72.63 | 83.97 | OOT | 81.15 | OOM | 82.15 | 84.04 |
| | SAGE | 50.43 | 87.27 | OOM | 87.34 | OOT | 87.34 | OOT | 87.42 | 73.57 | 86.68 | OOT | 87.34 | OOM | 83.45 | 88.88 |
| | GCN | 50.45 | 82.06 | OOM | 83.56 | OOT | 83.56 | OOT | 83.74 | 73.14 | 82.39 | OOT | 78.03 | OOM | 70.94 | 85.54 |
| | GIN | 51.82 | 83.13 | OOM | 84.31 | OOT | 84.07 | OOT | 82.93 | 73.15 | 83.51 | OOT | 82.85 | OOM | 80.72 | 86.50 |
| Phy. | GAT | 29.18 | 88.06 | OOM | 88.47 | OOT | 88.47 | OOT | 88.68 | 58.96 | 88.06 | OOT | 86.22 | OOM | 86.12 | 88.58 |
| | SAGE | 29.57 | 93.47 | OOM | 93.47 | OOT | 93.47 | OOT | 93.78 | 56.97 | 93.60 | OOT | 92.79 | OOM | 89.58 | 94.19 |
| | GCN | 27.84 | 91.27 | OOM | 91.08 | OOT | 91.08 | OOT | 91.78 | 58.35 | 91.07 | OOT | 84.85 | OOM | 88.46 | 91.48 |
| | GIN | 28.38 | 92.69 | OOM | 92.04 | OOT | 92.04 | OOT | 92.27 | 58.26 | 92.07 | OOT | 86.03 | OOM | 87.20 | 88.89 |

### 4.1 Computational Efficiency.

Existing methods like $k$-NN and $log$-model struggle to learn graph structures even for 20k nodes due to out-of-memory (OOM) or out-of-time (OOT) issues, while $l2$-model and $large$-model struggle beyond 50k nodes. Although $A$-NN and $emp$-Covar. are faster, GraphFLEx outperforms them on sufficiently large graphs (Table 2). While traditional methods may be efficient for small graphs, GraphFLEx scales significantly better, excelling on large datasets like *Pubmed* and *Syn 5*, where most methods fail. It accelerates structure learning, making $A$-NN 3× faster and $emp$-Covar. 2× faster.

### 4.2 Node Classification Accuracy

**Experimental Setup.** We now evaluate the prediction performance of GNN models when trained on graph structures learned from three distinct scenarios: **1) Original Structure:** GNN models trained

on the original graph structure, which we refer to as the Base Structure, **2) GraphFLEx Structure:** GNN models trained on the graph structure learned from GraphFLEx, and **3)Vanilla Structure:** GNN models trained on the graph structure learned from other existing methods. For each scenario, a unique graph structure is obtained. We trained GNN models on each of these three structure. For more details on GNN model parameters, see Appendix H.

**GNN Models.** Graph neural networks (GNNs) such as $GCN$ [51], $GraphSage$ [52], $GIN$ [53], and $GAT$ [54] rely on accurate message passing, dictated by the graph structure, for effective embedding. We use these models to evaluate the above-mentioned learned structures. Table 3 reports node classification performance across all methods. Notably, GraphFLEx outperforms vanilla structures by a significant margin across all datasets, achieving accuracies close to those obtained with the original structure. Figure 8 in Appendix H illustrates $GraphSage$ classification results, highlighting GraphFLEx's superior performance. For the $CS$ dataset, GraphFLEx ($large$-model) and GraphFLEx ($empCovar.$-model) even surpass the original structure, demonstrating its ability to preserve key structural properties while denoising edges, leading to improved accuracy.

### 4.3 Scalability of GraphFLEx on Large-Scale Graphs.

To comprehensively evaluate GraphFLEx's scalability to large-scale graphs, we consider four datasets with a high number of nodes: (a) *Flickr(89k nodes)* [55], (b) *Reddit (233k nodes)* [55], (c) *Ogbn-arxiv (169k nodes)* [46], and (d) *Ogbn-products (2.4M nodes)* [56]. As shown in Table 4, GraphFLEx consistently demonstrates superior scalability across all datasets, outperforming all baseline methods in runtime. In particular, methods such as *log-model*, *l2-model*, and *large-model* fail to run even on *Flickr*, while GraphFLEx successfully scales them on *Flickr*, *Ogbn-arxiv*, and *Reddit*, enabling structure learning where others cannot. For the most computationally demanding dataset, *Ogbn-products*, these methods remain prohibitively expensive even for GraphFLEx. Nonetheless, GraphFLEx efficiently supports scalable structure learning on *Ogbn-products* using the *Covar*, *ANN*, and *KNN* modules. Table 4 also reports node classification accuracy, demonstrating that GraphFLEx maintains performance comparable to the original (base) structure across all datasets. These results confirm that GraphFLEx not only scales effectively, but also preserves the quality of learned structures.

Table 4: Runtime (sec) and Node Classification Accuracy (%) across large datasets. Each cell shows: **Time / Accuracy**. Van = Vanilla, GFlex = GraphFLEx. OOM = Out of Memory, OOT = Out of Time.

| Method | ogbn-arxiv (60.13) | | ogbn-products (73.72) | | Flickr (44.92) | | Reddit (94.15) | |
|---|---|---|---|---|---|---|---|---|
| | Van. | GFlex | Van. | GFlex | Van. | GFlex | Van. | GFlex |
| Covar | OOM \| – | 3.7k \| 60.26 | OOM \| – | 83.1k \| 68.23 | 2.3k \| 44.65 | 682 \| 44.34 | OOM \| – | 6.6k \| 94.13 |
| ANN | 7.8k \| 60.14 | 4.8k \| 60.22 | OOM \| – | 89.3k \| 67.91 | 2.5k \| 44.09 | 705 \| 44.92 | 12.6k \| 94.14 | 6.1k \| 94.18 |
| knn | 8.3k \| 60.09 | 6.1k \| 60.23 | OOM \| – | 91.8k \| 68.47 | 2.7k \| 43.95 | 920 \| 44.73 | 15.6k \| 94.14 | 6.9k \| 94.15 |
| l2 | OOT \| – | 9.1k \| 58.39 | OOT \| – | OOT \| – | 93.3k \| 44.90 | 1.2k \| 44.32 | OOT \| – | 5.1 \| 93.47 |
| log | OOT \| – | 45.6k \| 58.72 | OOT \| – | OOT \| – | OOT \| – | 18.7k \| 44.59 | OOT \| – | 60.3k \| 94.13 |
| large | OOT \| – | 5.6k \| 60.20 | OOT \| – | OOT \| – | OOT \| – | 2.2k \| 44.45 | OOT \| – | 9.3k \| 93.71 |

### 4.4 GraphFLEx for Link Prediction and Graph Classification.

To further validate the generalization of our framework, we evaluate GraphFLEx on the link prediction task. The results are presented in Table 5, following the same setting as Table 3. The structure learned by GraphFLEx demonstrates strong predictive performance, in some cases even outperforming the base structure. This highlights the effectiveness of GraphFLEx in preserving and even enhancing relational information relevant for link prediction. While our primary focus is on structure learning

Table 5: Link predication accuracy (%) across different datasets. The best and the second-best accuracies in each row are highlighted by dark and lighter shades of Green, respectively.

| Data | ANN | | KNN | | log-model | | l2-model | | COVAR | | large-model | | Base Struct. |
|---|---|---|---|---|---|---|---|---|---|---|---|---|---|
| | Van. | GFlex | Van. | GFlex | Van. | GFlex | Van. | GFlex | Van. | GFlex | Van. | GFlex | |
| DBLP | 96.57 | 96.61 | OOM | 94.23 | OOT | 97.59 | OOT | 97.59 | 97.22 | 97.59 | OOT | 96.24 | 95.13 |
| Citeseer | 80.12 | 96.32 | 85.17 | 96.24 | 80.48 | 96.24 | 80.48 | 96.48 | 82.05 | 96.24 | 84.50 | 94.38 | 90.78 |
| Cora | 84.47 | 95.30 | 79.23 | 95.14 | 90.63 | 95.45 | 90.81 | 95.14 | 86.05 | 95.30 | 90.63 | 94.67 | 89.53 |
| Pubmed | 94.24 | 96.91 | OOM | 97.42 | OOT | 97.42 | OOT | 97.37 | 94.89 | 94.64 | OOT | 94.41 | 94.64 |
| CS | 94.21 | 95.73 | OOM | 96.02 | OOT | 93.17 | OOT | 93.17 | 93.52 | 92.31 | OOT | 95.73 | 95.00 |
| Physics | 95.77 | 91.34 | OOM | 94.63 | OOT | 90.79 | OOT | 94.63 | 92.03 | 90.79 | OOT | 92.97 | 93.96 |

for node-level tasks, we briefly discuss the applicability of GraphFLEx to graph classification. In such tasks, especially in domains like molecule or drug discovery, each data point often corresponds to a small individual subgraph. For these cases, applying clustering and coarsening is typically redundant and may introduce unnecessary computational overhead. Nevertheless, GraphFLEx

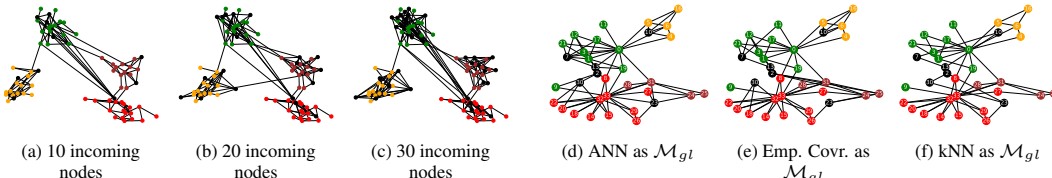

| (a) 10 incoming nodes | (b) 20 incoming nodes | (c) 30 incoming nodes | (d) ANN as $\mathcal{M}_{gl}$ | (e) Emp. Covr. as $\mathcal{M}_{gl}$ | (f) kNN as $\mathcal{M}_{gl}$ |

Figure 4: Figures (a), (b), and (c) illustrate the growing structure learned using GraphFLEx for *HE* synthetic dataset. Figures (d), (e), and (f) illustrate the learned structure on Zachary's karate dataset when existing methods are employed with GraphFLEx. New nodes are denoted using black color.

remains flexible—its learning module can be directly used without the clustering or coarsening steps, making it suitable for graph classification as well. This adaptability reinforces GraphFLEx's utility across a broad range of graph learning tasks.

### 4.5 Clustering Quality

We measure three metrics to evaluate the resulting clusters or community assignments: a) Normalized Mutual Information (NMI) [24] between the cluster assignments and original labels; b) Conductance ($\mathcal{C}$) [57] which measures the fraction of total edge volume that points outside the cluster; and c) Modularity ($\mathcal{Q}$) [49] which measures the divergence between the intra-community edges and the expected one. Table 6 illustrates these metrics for single-cell

Table 6: Clustering (NMI, $\mathcal{C}$, $\mathcal{Q}$) and node classification accuracy using GCN, GraphSAGE, GIN, and GAT.

| Data | NMI | $\mathcal{C}$ | $\mathcal{Q}$ | GCN | SAGE | GIN | GAT |
|------|-----|-----|-----|-----|------|-----|-----|
| Bar. M. | 0.716 | 0.057 | 0.741 | 91.2 | 96.2 | 95.1 | 94.9 |
| Seger. | 0.678 | 0.102 | 0.694 | 91.0 | 93.9 | 94.2 | 92.3 |
| Mura. | 0.843 | 0.046 | 0.706 | 96.9 | 97.4 | 97.5 | 96.4 |
| Bar. H. | 0.674 | 0.078 | 0.749 | 95.3 | 96.4 | 97.2 | 95.8 |
| Xin | 0.741 | 0.045 | 0.544 | 98.6 | 99.3 | 98.9 | 99.8 |
| MNIST | 0.677 | 0.082 | 0.712 | 92.9 | 94.5 | 94.9 | 82.6 |

RNA and the MNIST dataset (where the whole structure is missing), and Figure 12 in Appendix K shows the PHATE [58] visualization of clusters learned using GraphFLEx's clustering module $\mathcal{M}_{clust}$. We also train the aforementioned GNN models for the node classification task in order to illustrate the efficacy of the learned structures; the accuracy values presented in Table 6, clearly highlight the significance of the learned structures, as reflected by the high accuracy values.

### 4.6 Structure Visualization

We evaluate the structures generated by Graph-FLEx through visualizations on four small datasets: (i) MNIST [59], consisting of hand-written digit images, where Figure 5(a) shows that images of the same digit are mostly connected; (ii) Pre-trained GloVe embeddings [60] of English words, with Figure 5(b) revealing that frequently used words are closely connected; (iii) A synthetic $H.E$ dataset (see Appendix F), demonstrating GraphFLEx's ability to handle

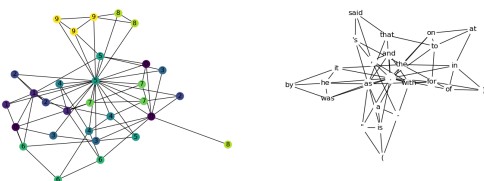

Figure 5: Effectiveness of our framework in learning structure between similar MNIST digits and GloVe embeddings.

expanding networks without requiring full relearning. Figure 4(a-c) shows the graph structure evolving as 30 new nodes are added over three timestamps; and (iv) Zachary's karate club network [61], which highlights GraphFLEx's multi-framework capability. Figure 4(d-f) shows three distinct graph structures after altering the learning module. For a comprehensive ablation study, refer to Appendix L.

## 5 Conclusion

Large or expanding graphs challenge the best of graph learning approaches. GraphFLEx, introduced in this paper, seamlessly adds new nodes into an existing graph structure. It offers diverse methods for acquiring the graph's structure. GraphFLEx consists of three key modules: Clustering, Coarsening, and Learning which empowers GraphFLEx to serves as a comprehensive framework applicable individually for clustering, coarsening, and learning tasks. Empirically, GraphFLEx outperforms state-of-the-art baselines, achieving up to 3× speedup while preserving structural quality. It achieves accuracies close to training on the original graph, in most instances. The performance across multiple real and synthetic datasets affirms the utility and efficacy of GraphFLEx for graph structure learning.

**Limitations and Future Work.** GraphFLEx is designed assuming minimal inter-community connectivity, which aligns well with many real-world scenarios. However, its applicability to heterophilic graphs may require further adaptation. Future work will focus on extending the framework to supervised GSL methods and heterophilic graphs, broadening its scalability and versatility.

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

# Appendix

## A  Degree-Corrected Stochastic Block Model(DC-SBM)

The DC-SBM is one of the most commonly used models for networks with communities and postulates that, given node labels $\mathbf{c} = c_1, ...c_n$, the edge variables $A'_{ij}s$ are generated via the formula

$$E[A_{ij}] = \theta_i \theta_j P_{c_i} P_{c_j}$$

, where $\theta_i$ is a "degree parameter" associated with node $i$, reflecting its individual propernsity to form ties, and $P$ is a $K \times K$ symmetric matrix containing the between/withincommunity edge probabilities and $P_{c_i} P_{c_j}$ denotes the edge probabilities between community $c_i$ and $c_j$.
For DC-SBM model [50] assumed $P_n$ on $n$ nodes with $k$ classes, each node $v_i$ is given a label/degree pair$(c_i, \theta_i)$, drawn from a discrete joint distribution $\Pi_{K \times m}$ which is fixed and does not depend on n. This implies that each $\theta_i$ is one of a fixed set of values $0 \leq x_1 \leq .... \leq x_m$. To facilitate analysis of asymptotic graph sparsity, we parameterize the edge probability matrix $P$ as $P_n = \rho_n P$ where P is independent of $n$, and $\rho_n = \lambda_n/n$ where $\lambda_n$ is the average degree of the network.

## B  Neighbourhood Preservation

**Theorem 2.** *Neighborhood Preservation. Let $\mathcal{N}_k(\mathcal{E}_i)$ denote the neighborhood of incoming nodes $\mathcal{E}_i$ for the $i^{th}$ community. With partition matrix $\mathcal{P}_i$ and $\mathcal{M}_{gl}(S_i, X^i_\tau) = \mathcal{G}^c_\tau(V^c_\tau, A^c_\tau)$ we identify the supernodes connected to incoming nodes $\mathcal{E}_i$ and subsequently select nodes within those supernodes; this subset of nodes is denoted by $\omega_{V^i_\tau}$. Formally,*

$$\omega_{V^i_\tau} = \bigcup_{v \in \mathcal{E}_i} \left\{ \bigcup_{s \in S_i} \{\pi^{-1}(s) | A^c_\tau(v, s) \neq 0\} \right\}$$

*Then, with probability $\Pi_{\{c \in \phi\}} p(c)$, it holds that $\mathcal{N}_k(\mathcal{E}_i) \subseteq \omega_{V^i_\tau}$ where*

$$p(c) \leq 1 - \frac{2}{\sqrt{2\pi}} \frac{c}{r} \left[ 1 - e^{-r^2/(2c^2)} \right],$$

*and $\phi$ is a set containing all pairwise distance values $(c = \|v - u\|)$ between every node $v \in \mathcal{E}_i$ and the nodes $u \in \omega_{V^i_\tau}$. Here, $\pi^{-1}(s)$ denotes the set of nodes mapped to supernode $s$, $r$ is the bin-width hyperparameter of $\mathcal{M}_{coar}$.*

**Proof:** The probability that LSH random projection [32, 62] preserves the distance between two nodes $v$ and $u$ i.e., $d(u, v) = c$, is given by:

$$p(c) = \int_0^r \frac{1}{c} f_2\left(\frac{t}{c}\right) \left(1 - \frac{t}{r}\right) dt,$$

where $f_2(x) = \frac{2}{\sqrt{2\pi}} e^{-x^2/2}$ represents the Gaussian kernel when the projection matrix is randomly sampled from $p$-stable$(p = 2)$ distribution [62].
The probability $p(c)$ can be decomposed into two terms:

$$p(c) = S_1(c) - S_2(c),$$

$S_1(c)$ and $S_2(c)$ are defined as follows:

$$S_1(c) = \frac{2}{\sqrt{2\pi}} \int_0^r e^{-(t/c)^2/2} dt \leq 1,$$

$$S_2(c) = \frac{2}{\sqrt{2\pi}} \int_0^r e^{-(t/c)^2/2} \frac{t}{r} dt.$$

$$S_2(c) = \frac{2}{\sqrt{2\pi}} \cdot \frac{c}{r} \int_0^r e^{-(t/c)^2/2} \frac{t}{c^2} dt$$

Expanding $S_2(c)$ :

$$S_2(c) = \frac{2}{\sqrt{2\pi}} \cdot \frac{c}{r} \int_0^{r^2/(2c^2)} e^{-y} dy$$

$$S_2(c) = \frac{2}{\sqrt{2\pi}} \cdot \frac{c}{r} \left[ 1 - e^{-r^2/(2c^2)} \right]$$

Thus, the probability $p(c)$ can be bounded as:

$$p(c) \le 1 - \frac{2}{\sqrt{2\pi}} \frac{c}{r} \left[ 1 - e^{-r^2/(2c^2)} \right].$$

Now, let $\phi$ be the set of all pairwise distances $d(u, v)$, where $v \in \mathcal{E}_i$ and node$\omega_{V_\tau^i}$. The probability that all nodes in $\mathcal{N}_k(\mathcal{E}_i)$ are preserved within $\omega_{V_\tau^i}$, requires that all distances $c \in \phi$ are also preserved. The probability is then given by:

$$\prod_{c \in \phi} p(c).$$

$$\prod_{c \in \phi} p(c) \le \prod_{c \in \phi} \left( 1 - \frac{2}{\sqrt{2\pi}} \frac{c}{r} \left[ 1 - e^{-r^2/(2c^2)} \right] \right).$$

## C    Continual Learning and Dynamic Graph Learning

In this subsection, we highlight the key distinctions between Graph Structure Learning (GSL) and related fields to justify our specific selection of related works in Section 2.2. GSL is often confused with topics such as Continual Learning (CL) and Dynamic Graph Learning (DGL).

CL [39–41] addresses the issue of catastrophic forgetting, where a model's performance on previously learned tasks degrades significantly after training on new tasks. In CL, the model has access only to the current task's data and cannot utilize data from prior tasks. Conversely, DGL [42–44] focuses on capturing the evolving structure of graphs and maintaining updated graph representations, with access to all prior information.

While both *CL and DGL* aim to *enhance model adaptability* to dynamic data, GSL is primarily concerned with generating *high-quality graph structures* that can be leveraged for downstream tasks such as node classification [51], link prediction [63], and graph classification [64]. Moreover, in CL and DGL, different tasks typically involve distinct data distributions, whereas GSL assumes a consistent data distribution throughout.

## D    Related Work

Table 7 presents the formulations and associated time complexities of various unsupervised Graph Structure Learning methods.

Table 7: Unsupervised Graph Structure Learning Methods

| Method | Time Complexity | Formulation |
|--------|-----------------|-------------|
| $GLasso$ | $O(N^3)$ | $\max_\Theta \log \det \Theta - \mathrm{tr}(\hat{\Sigma}\Theta) - \rho\|\Theta\|_1$ |
| $log$-model | $O(N^2)$ | $\min_{W \in \mathcal{W}} \|W \circ Z\|_{1,1} - \alpha \mathbf{1}^T \log(W\mathbf{1}) + \frac{\beta}{2}\|W\|_F^2$ |
| $l2$-model | $O(N^2)$ | $\min_{W \in \mathcal{W}} \|W \circ Z\|_{1,1} + \alpha\|W\mathbf{1}\|^2 + \alpha\|W\|_F^2 + \mathbf{1}\{\|W\|_{1,1} = n\}$ |
| $large$-model | $O(N \log(N))$ | $\min_{W \in \tilde{W}} \|W \circ Z\|_{1,1} - \alpha \mathbf{1}^T \log(W\mathbf{1}) + \frac{\beta}{2}\|W\|_F^2$ |

## E    Run Time Analysis

In the context of clustering module, $k - NN$ is the fastest algorithm, while Spectral Clustering is the slowest. Suppose we aim to learn the structure of a graph with $N$ nodes. The clustering module, however, is only applied to a randomly sampled, smaller, static subgraph with $k$ nodes, where $k \ll N$. In the worst-case scenario, spectral clustering requires $\mathcal{O}(k^3)$ time, whereas in the best case, $k - NN$ requires $\mathcal{O}(k^2)$ time. For coarsening module, LSH-based coarsening framework [30], has the best time complexity of $\mathcal{O}(\frac{k_\tau}{c})$ while FGC denotes the worst case with a time-complexity of $\mathcal{O}((\frac{k_\tau}{c})^2\|S_\tau^i\|)$ where $c$ is the number of communities detected by clustering module $\mathcal{M}_{\mathrm{clust}}$, $\|S_\tau^i\|$ is the number of coarsened node in the relevant community at $\tau$ timestamp and $k_\tau$ denotes number of nodes at $\tau$ timestamp. For learning module, $A - NN$ is the most efficient algorithm with time

complexity as $\mathcal{O}(NlogN)$, while $GLasso$ has the worst computational cost of $\mathcal{O}(N^3)$. So, the effective time complexity of GraphFLEx is upper bounded by $\mathcal{O}(k^3 + (\frac{k_\tau}{c})^2\|S_\tau^i\| + \alpha^3)$ and lower bounded by $\mathcal{O}(k^2 + \frac{k_\tau}{c} + \alpha log\alpha)$ where $\alpha = \|S_\tau^i\| + \|\mathcal{E}_\tau^i\|$. GraphFLEx's efficiency in term of computational time is evident in Figure 1 and further quantified in Table 2.

Out of the three modules of GraphFLEx first module($\mathcal{M}_{\text{clust}}$) is trained once, and hence its run time is always bounded; computational time for second module($\mathcal{M}_{\text{coar}}$) can also be controlled because some of the methods either needs training once [65] or have linear time complexity [30]. Consequently, both the clustering and coarsening modules contribute linearly to the overall time complexity, denoted as $\mathcal{O}(N)$. Thus, the effective time complexity of GraphFLEx is given by $\mathcal{O}(N + \mathcal{O}(\mathcal{M}_{gl}(\|S_i, X_\tau^i\|))$. The overall complexity scales either linearly or sub-linearly, depending on $\alpha$ and the $\mathcal{M}_{gl}$ module. For instance, when $\mathcal{M}_{gl}$ is A-NN the complexity remains linear, if $\alpha \log(\alpha) \approx N$, whereas for $GLasso$, a linear behavior is observed when $\alpha^3 \approx N$.

# F Datasets

Datasets used in our experiments vary in size, with nodes ranging from 1k to 60k. Table 8 lists all the datasets we used in our work. We evaluate our proposed framework $GraphFlex$ on real-world datasets *Cora ,Citeseer, Pubmed* [66], *CS, Physics* [67], *DBLP* [68], all of which include graph structures. These datasets allow us to compare the learned structures with the originals. Additionally, we utilize single-cell RNA pancreas datasets [69], including Baron, Muraro, Segerstolpe, and Xin, where the graph structure is missing. The Baron dataset was downloaded from the Gene Expression Omnibus (GEO) (accession no. GSE84133). The Muraro dataset was downloaded from GEO (accession no. GSE85241). The Segerstolpe dataset was accessed from ArrayExpress (accession no. E-MTAB-5061). The Xin dataset was downloaded from GEO (accession no. GSE81608). We simulate the expanding graph scenario by splitting the original dataset across different $\mathcal{T}$ timestamps. We assumed 50% of the nodes were static, with the remaining nodes arriving as incoming nodes at different timestamps.

**Synthetic datasets:** Different data generation techniques validate that our results are generalized to different settings. Please refer to Table 8 for more details about the number of nodes, edges, features, and classes, $Syn$ denotes the type of synthetic datasets. Figure 6 shows graphs generated using different methods. We have employed three different ways to generate synthetic datasets which are mentioned below:

- **PyGSP(PyGsp):** We used synthetic graphs created by PyGSP [70] library. PyG-G and PyG-S denotes grid and sensor graphs from PyGSP.
- **Watts–Strogatz's small world(SW):** [71] proposed a generation model that produces graphs with small-world properties, including short average path lengths and high clustering.
- **Heterophily(HE):** We propose a method for creating synthetic datasets to explore graph behavior across a heterophily spectrum by manipulating heterophilic factor $\alpha$, and classes. $\alpha$ is determined by dividing the number of edges connecting nodes from different classes by the total number of edges in the graph.

**Visulization Datasets:** To evaluate, the learned graph structure, we have also included three datasets: (i) MNIST [59], consisting of handwritten digit images; (ii) Pre-trained GloVe embeddings [60] of English words; and (iii) Zachary's karate club network [61].

**Large Datasets:** To comprehensively evaluate GraphFLEx's scalability to large-scale graphs, we consider four datasets with a high number of nodes: (a) *Flickr(89k nodes)* [55], (b) *Reddit (233k nodes)* [55], (c) *Ogbn-arxiv (169k nodes)* [46], and (d) *Ogbn-products (2.4M nodes)* [56].

*System Specifications:* All the experiments conducted for this work were performed on an Intel Xeon W-295 CPU with 64GB of RAM desktop using the Python environment.

| Category | Data | Nodes | Edges | Feat. | Class | Type |
|---|---|---|---|---|---|---|
| Original Structure Known | Cora | 2,708 | 5,429 | 1,433 | 7 | Citation network |
| | Citeseer | 3,327 | 9,104 | 3,703 | 6 | Citation network |
| | DBLP | 17,716 | 52.8k | 1,639 | 4 | Research paper |
| | CS | 18,333 | 163.7k | 6,805 | 15 | Co-authorship network |
| | PubMed | 19,717 | 44.3k | 500 | 3 | Citation network |
| | Physics | 34,493 | 247.9k | 8,415 | 5 | Co-authorship network |
| Original Structure Not Known | Xin | 1,449 | NA | 33,889 | 4 | Human Pancreas |
| | Baron Mouse | 1,886 | NA | 14,861 | 13 | Mouse Pancreas |
| | Muraro | 2,122 | NA | 18,915 | 9 | Human Pancreas |
| | Segerstolpe | 2,133 | NA | 22,757 | 13 | Human Pancreas |
| | Baron Human | 8,569 | NA | 17,499 | 14 | Human Pancreas |
| Synthetic | Syn 1 | 2,000 | 8,800 | 150 | 4 | SW |
| | Syn 2 | 5,000 | 22k | 150 | 4 | SW |
| | Syn 3 | 10,000 | 44k | 150 | 7 | SW |
| | Syn 4 | 50,000 | 220k | 150 | 7 | SW |
| | Syn 5 | 400 | 1,520 | 100 | 4 | PyG-G |
| | Syn 6 | 2,500 | 9,800 | 100 | 4 | PyG-S |
| | Syn 7 | 1,000 | 9,990 | 150 | 4 | HE |
| | Syn 8 | 2,000 | 40k | 150 | 4 | HE |
| Visulization Datasets | MNIST | 60,000 | NA | 784 | 10 | Images |
| | Zachary's karate | 34 | 156 | 34 | 4 | Karate club network |
| | Glove | 2,000 | NA | 50 | NA | GloVe embeddings |
| Large dataset | Flickr | 89,250 | 899,756 | 500 | 7 | - |
| | Reddit | 232,965 | 11.60M | 602 | 41 | - |
| | Ogbn-arxiv | 169,343 | 1.16M | 128 | 40 | - |
| | Ogbn-products | 2,449,029 | 61.85M | 100 | 47 | - |

Table 8: Summary of the datasets.

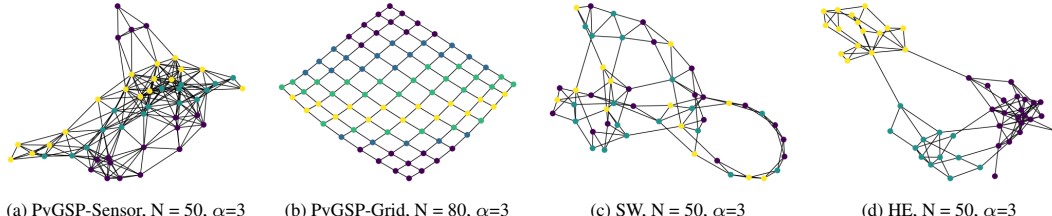

(a) PyGSP-Sensor, N = 50, $\alpha$=3    (b) PyGSP-Grid, N = 80, $\alpha$=3    (c) SW, N = 50, $\alpha$=3    (d) HE, N = 50, $\alpha$=3

Figure 6: This figure illustrates different types of synthetic graphs generated using i)PyGSP, ii) Watts–Strogatz's small world(SW), and iii) Heterophily(HE). N denotes the number of nodes, while $\alpha$ denotes the number of classes.

# G   Algorithm

---

**Algorithm 1** GraphFlex: A Unified Structure Learning framework for expanding and Large Scale Graphs

---

**Input**: Graph $G_0(X_0, A_0)$, expanding nodes set $\mathcal{E}_1^T = \{\mathcal{E}_\tau(\mathcal{V}_\tau, \mathcal{X}_\tau)\}_{\tau=1}^T$
**Parameter**: GClust, GCoar, GL $\leftarrow$ Clustering, Coarsening and Learning Module
**Output**: Graph $G_T(X_T, A_T)$

1: Train clustering module $train(\mathcal{M}_{clust}, \text{GClust}, G_0)$
2: **for** each $E_t(V_t, X_t)$ in $\mathcal{E}_1^T$ **do**
3:    $C_t = infer(\mathcal{M}_{clust}, X_t), C_t \in \mathbb{R}^{N_t}$ denotes the communities of $N_t$ nodes at time $t$.
4:    $I_t = unique(C_t)$.
5:    **for** each $I_t^i$ in $I_t$ **do**
6:       $G_{t-1}^i = \text{subgraph}(G_{t-1}, I_t^i)$
7:       $\{S_{t-1}^i, P_{t-1}^i\} = \mathcal{M}_{coar}(G_{t-1}^i), S_{t-1}^i \in \mathbb{R}^{k \times d}$ are features of $k$ supernodes, $P_{t-1}^i \in \mathbb{R}^{k \times N_t^i}$ is the partition matrix.
8:       $Gc_{t-1}^i(S_{t-1}^i, A_{t-1}^i) = \mathcal{M}_{gl}(S_{t-1}^i, X_t^i), Gc_{t-1}^i$ is the learned graph on super-nodes $S_{t-1}^i$ and new node $X_t^i$.
9:       $\omega_t^i \leftarrow \texttt{[]}$
10:      **for** $x \in X_t^i$ **do**
11:        $\omega_t^i.append(x)$
12:        $n_p = \{n \mid A_{t-1}^i[n] > 0\}$
13:        $\omega_t^i.append(n_p)$
14:      **end for**
15:      $G_{t-1} = update(G_{t-1}, \mathcal{M}_{gl}(\omega_t^i))$
16:    **end for**
17:    $G_t = G_{t-1}$
18: **end for**
19: **return** $G_T(X_T, A_T)$

---

# H   Other GNN models

We used four GNN models, namely GCN, GraphSage, GIN, and GAT. Table 9 contains parameter details we used to train GraphFlex. We have used these parameters across all methods.

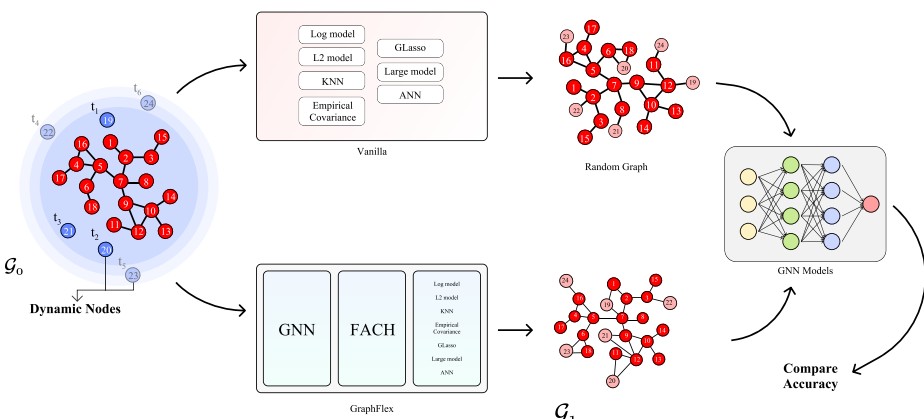

Figure 7: GNN training pipeline.

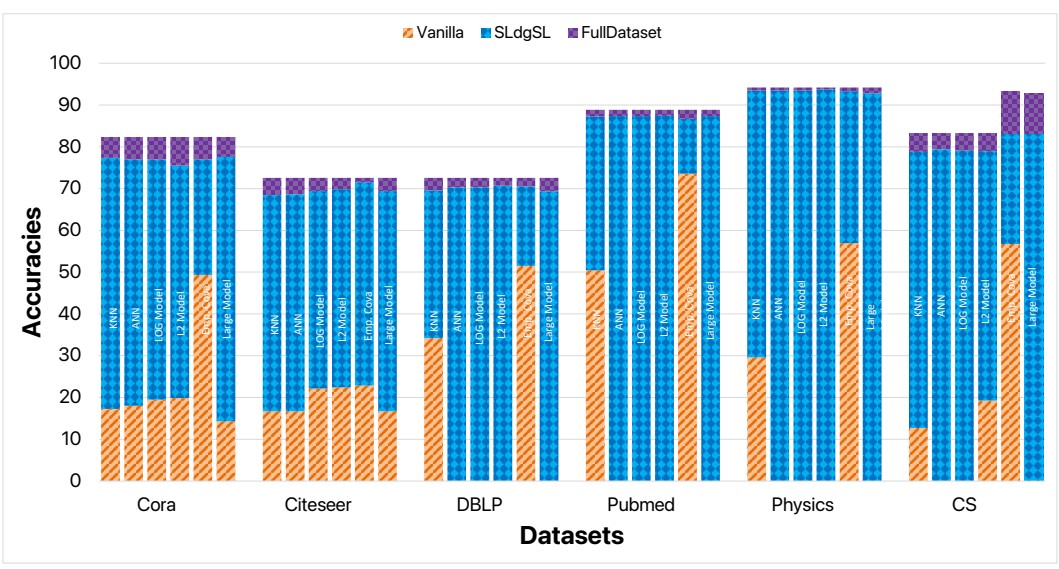

Figure 8: GraphSage accuracies when structure is learned or given with 3 different scenarios(Vanilla, GraphFlex, Original) across different datasets, highlighting performance with 30% node growth over 25 timestamps.

Figure 7 illustrates the pipeline for training our GNN models. Graph structures were learned using both existing methods and GraphFlex, and GNN models were subsequently trained on both structures. Results across all datasets are presented in Table 10 and Table 3.

Table 9: GNN model parameters.

| Model | Hidden Layers | L.R | Decay | Epoch |
|---|---|---|---|---|
| GCN | $\{64, 64\}$ | 0.003 | 0.0005 | 500 |
| GraphSage | $\{64, 64\}$ | 0.003 | 0.0005 | 500 |
| GIN | $\{64, 64\}$ | 0.003 | 0.0005 | 500 |
| GAT | $\{64, 64\}$ | 0.003 | 0.0005 | 500 |

We randomly split data in 60%, 20%, 20% for training-validation-test. The results for these models on synthetic datasets are presented in Table 10.

Figure 7 illustrates the pipeline for training our GNN models. Graph structures were learned using both existing methods and GraphFlex, and GNN models were subsequently trained on both structures.

## I   Computational Efficiency

Table 11 illustrates the remaining computational time for learning graph structures using GraphFLEx with existing Vanilla methods on Synthetic datasets. While traditional methods may be efficient for small graphs, GraphFLEx scales significantly better, excelling on large datasets like *Pubmed* and *Syn 5*, where most methods fail.

## J   Visualization of Growing graphs

This section helps us visualize the phases of our growing graphs. We have generated a synthetic graph of 60 nodes using PyGSP-Sensor and HE methods mentioned in Appendix F. We then added 40 new nodes denoted using black color in these existing graphs at four different timestamps. Figure 9 and Figure 10 shows the learned graph structure after each timestamp for two different Synthetic graphs.

## K   Clustering Quality

Figure 12 shows the PHATE [58] visualization of clusters learned using GraphFLEx's clustering module $\mathcal{M}_{clust}$ for 6 single-cell RNA datasets, namely $Xin$, $MNIST$, $Baron - Human$, $Muraro$, $BaronMouse$, and $Segerstolpe$ datasets.

Table 10: Node classification accuracies on different GNN models using GraphFLEx (GFlex) with existing Vanilla (Van.) methods. The experimental setup involves treating 70% of the data as static, while the remaining 30% of nodes are treated as new nodes coming in 25 different timestamps. The best and the second-best accuracies in each row are highlighted by dark and lighter shades of Green, respectively. GraphFLEx's structure beats all of the vanilla structures for every dataset. OOM and OOT denotes out-of-memory and out-of-time respectively.

| Dataset | Model | ANN | | KNN | | log-model | | l2-model | | COVAR | | large-model | | Base Struc. |
|---|---|---|---|---|---|---|---|---|---|---|---|---|---|---|
| | | Van. | GFlex | Van. | GFlex | Van. | GFlex | Van. | GFlex | Van. | GFlex | Van. | GFlex | |
| Cora | GAT | 18.73 | 73.84 | 20.96 | 73.65 | 16.14 | 72.36 | 18.74 | 73.10 | 49.72 | 77.55 | 14.28 | 76.43 | 79.77 |
| | SAGE | 17.25 | 77.37 | 18.00 | 76.99 | 19.48 | 77.40 | 19.85 | 75.51 | 49.35 | 76.99 | 14.28 | 77.55 | 82.37 |
| | GCN | 17.99 | 78.11 | 17.81 | 77.92 | 18.55 | 77.74 | 20.41 | 79.22 | 47.31 | 80.52 | 14.28 | 79.03 | 84.60 |
| | GIN | 16.69 | 76.44 | 18.74 | 80.52 | 17.44 | 76.25 | 19.29 | 76.62 | 48.79 | 78.85 | 14.28 | 76.06 | 81.63 |
| Citeseer | GAT | 16.51 | 61.82 | 25.00 | 62.27 | 19.24 | 64.70 | 18.18 | 63.48 | 20.91 | 62.73 | 16.67 | 62.27 | 66.42 |
| | SAGE | 16.66 | 68.48 | 16.67 | 68.64 | 22.12 | 69.39 | 22.42 | 69.85 | 22.88 | 71.52 | 16.67 | 69.39 | 72.57 |
| | GCN | 28.18 | 60.00 | 16.67 | 61.97 | 20.45 | 65.45 | 19.70 | 64.24 | 21.06 | 64.70 | 16.67 | 63.18 | 68.03 |
| | GIN | 16.66 | 64.39 | 16.67 | 63.94 | 20.15 | 59.85 | 18.64 | 63.64 | 22.12 | 60.30 | 16.67 | 61.81 | 67.38 |
| Syn 4 | GAT | 29.55 | 92.07 | OOM | 90.86 | OOT | 91.64 | OOT | 91.64 | 35.79 | 92.52 | OOT | 93.74 | 89.49 |
| | SAGE | 26.75 | 87.89 | OOM | 91.05 | OOT | 86.64 | OOT | 86.64 | 32.92 | 90.44 | OOT | 86.01 | 90.03 |
| | GCN | 28.85 | 51.97 | OOM | 19.58 | OOT | 18.29 | OOT | 18.92 | 33.80 | 26.60 | OOT | 36.85 | 21.43 |
| | GIN | 28.50 | 65.61 | OOM | 31.06 | OOT | 26.51 | OOT | 26.56 | 34.03 | 46.40 | OOT | 47.10 | 29.35 |
| Syn 6 | GAT | 44.00 | 86.80 | 43.60 | 86.60 | 30.00 | 78.75 | 55.40 | 92.80 | 36.20 | 93.60 | 31.80 | 92.80 | 97.20 |
| | SAGE | 41.00 | 93.80 | 41.40 | 93.60 | 33.75 | 88.75 | 57.60 | 94.00 | 35.20 | 94.80 | 28.20 | 95.60 | 97.40 |
| | GCN | 43.60 | 88.80 | 42.20 | 87.40 | 26.25 | 81.25 | 55.60 | 92.40 | 31.40 | 94.40 | 25.20 | 94.00 | 99.40 |
| | GIN | 39.60 | 89.00 | 40.40 | 86.60 | 21.25 | 82.50 | 55.20 | 91.80 | 30.00 | 94.60 | 30.40 | 92.00 | 98.80 |
| Syn 8 | GAT | 29.55 | 99.75 | 33.75 | 88.75 | 88.25 | 99.25 | 88.25 | 99.25 | 26.00 | 85.50 | 94.00 | 96.00 | 98.50 |
| | SAGE | 26.75 | 100.0 | 32.50 | 100.0 | 88.75 | 99.50 | 88.75 | 99.50 | 26.75 | 100.0 | 92.50 | 100.0 | 100.0 |
| | GCN | 28.85 | 98.75 | 31.75 | 99.75 | 88.75 | 99.00 | 88.75 | 99.00 | 28.50 | 99.25 | 95.00 | 100.0 | 100.0 |
| | GIN | 28.50 | 50.00 | 30.50 | 91.00 | 82.25 | 91.50 | 82.25 | 91.50 | 27.25 | 81.75 | 91.75 | 92.25 | 78.25 |

Table 11: Computational time for learning graph structures using GraphFLEx (GFlex) with existing methods (Vanilla referred to as Van.). The experimental setup involves treating 50% of the data as static, while the remaining 50% of nodes are treated as incoming nodes arriving in 25 different timestamps. The best times are highlighted by color Green. OOM and OOT denote out-of-memory and out-of-time, respectively.

| Data | ANN | | KNN | | log-model | | l2-model | | COVAR | | large-model | |
|---|---|---|---|---|---|---|---|---|---|---|---|---|---|
| | Van. | GFlex | Van. | GFlex | Van. | GFlex | Van. | GFlex | Van. | GFlex | Van. | GFlex |
| Syn 1 | 19.4 | 9.8 | 2.5 | 10.5 | 2418 | 56.4 | 37.2 | 8.8 | 3.5 | 8.3 | 205 | 9.4 |
| Syn 2 | 47.3 | 16.9 | 6.6 | 18.3 | 14000 | 144 | 214 | 22.6 | 20.3 | 18.6 | 1259 | 16.4 |
| Syn 5 | 5.1 | 11.5 | 0.8 | 7.3 | 57.4 | 28 | 1.1 | 5.8 | 0.2 | 4.8 | 3.2 | 5.3 |
| Syn 6 | 16.6 | 9.9 | 2.8 | 11.4 | 1766 | 96.3 | 193 | 101 | 5.3 | 8.9 | 324 | 9.6 |
| Syn 7 | 10.6 | 7.4 | 1.4 | 8.9 | 704 | 85.2 | 10.3 | 7.9 | 0.9 | 6.4 | 36.5 | 8.2 |
| Syn 8 | 19.6 | 11.2 | 2.5 | 11.7 | 2416 | 457 | 37.2 | 17.0 | 3.4 | 10.9 | 204 | 11.7 |

**PyGsp**

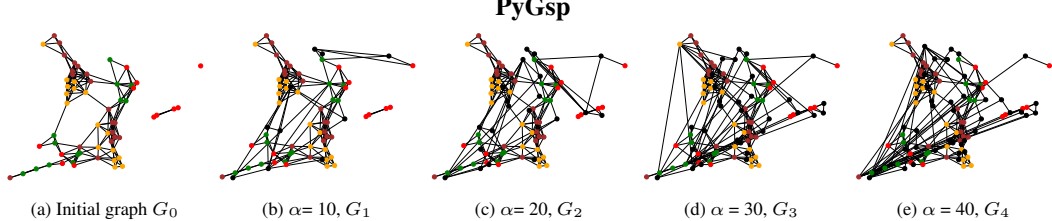

(a) Initial graph $G_0$    (b) $\alpha = 10$, $G_1$    (c) $\alpha = 20$, $G_2$    (d) $\alpha = 30$, $G_3$    (e) $\alpha = 40$, $G_4$

Figure 9: This figure illustrates the growing structure learned using GraphFlex for dynamic nodes. New nodes are denoted using black color, and $\alpha$ denotes number of new nodes. *PyGsp* denotes type synthetic graph.

## L  Ablation Study

In this section, we present an ablation study to analyze the role of individual modules within GraphFLEx and their influence on the final graph structure. Specifically, we focus on two aspects: (i) the significance of the clustering module, and (ii) the effect of varying module configurations on the learned graph topology.

**HE**

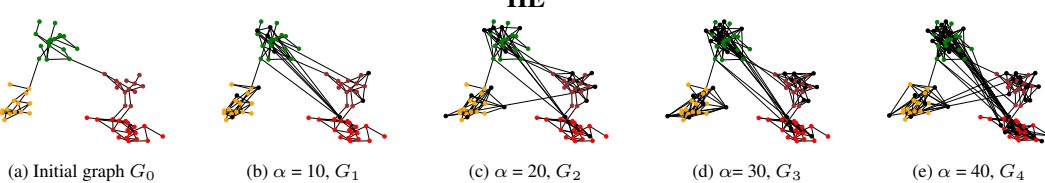

(a) Initial graph $G_0$   (b) $\alpha = 10$, $G_1$   (c) $\alpha = 20$, $G_2$   (d) $\alpha = 30$, $G_3$   (e) $\alpha = 40$, $G_4$

Figure 10: This figure illustrates the growing structure learned using GraphFlex for dynamic nodes. New nodes are denoted using black color, and $\alpha$ denotes the number of new nodes. *HE* denotes the type of synthetic graph.

*Original Karate Graph*

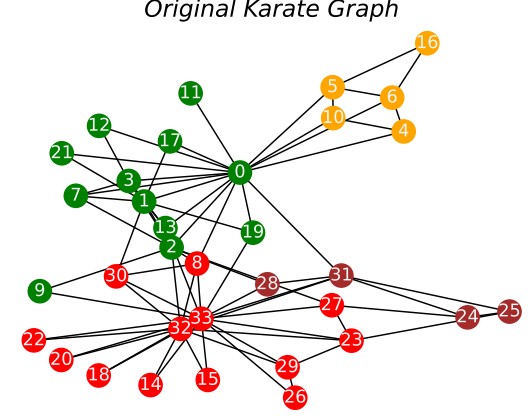

Figure 11: Original Karate Graph

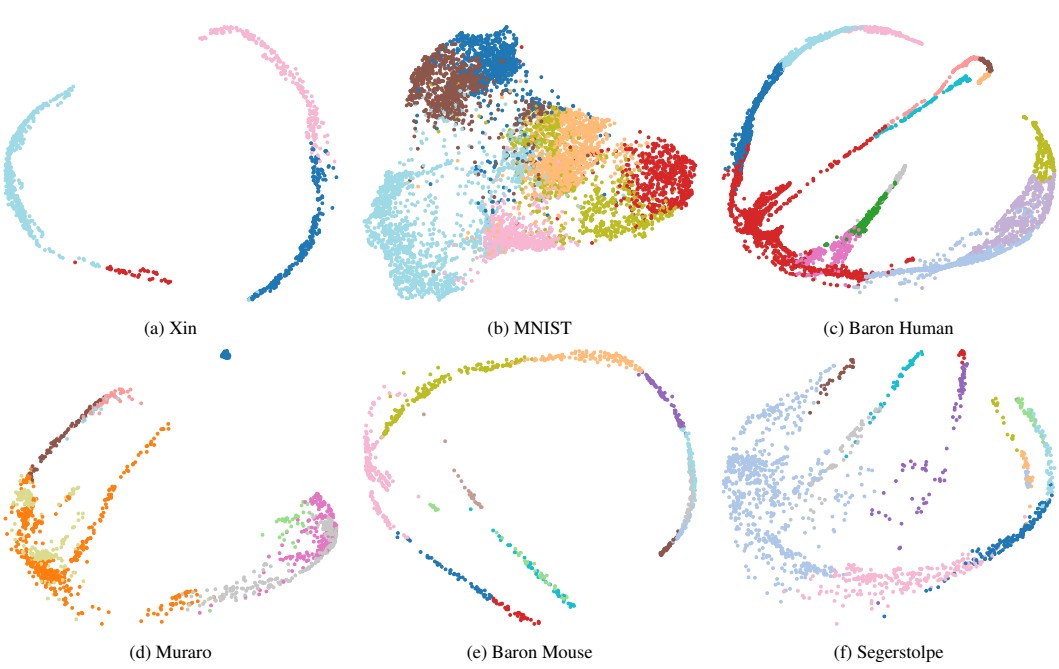

(a) Xin   (b) MNIST   (c) Baron Human

(d) Muraro   (e) Baron Mouse   (f) Segerstolpe

Figure 12: PHATE visualization of clusters learnt using GraphFlex clustering module for scRNA-seq datasets.

## L.1 Clustering Module Evaluation

To evaluate the effectiveness of the clustering module, we compute standard metrics such as Normalized Mutual Information (NMI), Conductance (C), and Modularity (Q) across various datasets (see Table 6 in Section 4.5). These metrics collectively validate the quality of the discovered clusters, thereby justifying the use of a clustering module as a foundational step in GraphFLEx. Since clustering in GraphFLEx is applied only once on a randomly sampled small set of nodes, selecting the right

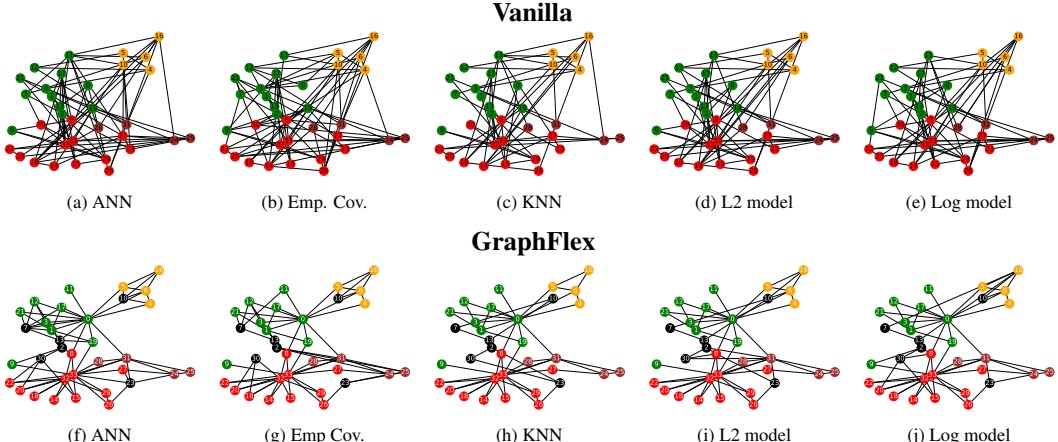

Figure 13: This figure compares the structures learned on Zachary's karate dataset when existing methods are employed with GraphFlex and when existing methods are used individually. We consider six nodes, denoted in black, as dynamic nodes.

method can be considered as part of hyperparameter tuning, where these clustering measures can guide the optimal choice based on dataset characteristics.

## L.2 Impact of Module Choices on Learned Graph Structure

This section involves a comparison of the graph structure learned from GraphFlex with existing methods. Six nodes were randomly selected and considered as new nodes. Figure 13 visually depicts the structures learned using GraphFlex compared to other methods. It is evident from the figure that the structure known with GraphFlex closely resembles the original graph structure. Figure 11 shows the original structure of Zachary's karate club network [61]. We assumed six random nodes to be dynamic nodes, and the structure learned using GraphFlex compared to existing methods is shown in Figure 13.

