# OpenReview forum: "GraphFLEx: Structure Learning $\underline{\text{F}}$ramework for $\underline{\text{L}}$arge $\underline{\text{Ex}}$panding $\underline{\text{Graph}}$s"
_NeurIPS.cc/2025/Conference — Submitted to NeurIPS 2025_

### Official Review · Reviewer_hvJy · 2025-07-03

**Clarity:** 4
**Significance:** 2
**Originality:** 3
**Rating:** 4
**Confidence:** 3

**Summary:**

This paper proposes GraphFLEx, a scalable and modular framework for unsupervised graph structure learning in large and dynamically expanding graphs. By combining graph clustering, coarsening, and structure learning, GraphFLEx enables efficient incremental updates and supports multiple method configurations. Experiments on 26 datasets demonstrate its advantages in scalability, runtime, and predictive performance compared to existing methods.

**Questions:**

1. In Table 2, GraphFLEx performs slower than vanilla methods on small datasets. Could the authors explain the source of this overhead? Which modules (clustering, coarsening, etc.) contribute most to the added cost?

2. How should users choose the partitioning ratio r and number of timestamps T? Are there any guidelines or ablation results that indicate their impact on final accuracy and efficiency?

3. In some cases, GraphFLEx achieves performance even better than the original graph structure. Could the authors explain whether this is due to denoising effects or improved structure inference? What conditions enable such gains?

4. The method relies heavily on the assumption of community homophily. How would the method perform under heterophilic settings, where inter-community links are common? Is there a way to relax this assumption?

5. Given the framework’s flexibility, is there a practical recommendation on how to select clustering, coarsening, and learning components for new datasets? Some empirical guidelines would be very useful.

**Ethical Concerns:**

["NO or VERY MINOR ethics concerns only"]

**Final Justification:**

see  above

**Limitations:**

Partially addressed. The authors briefly acknowledge that the method may not generalize well to heterophilic graphs due to the DC-SBM assumption. However, this point should be discussed more explicitly, possibly with experimental illustration. Furthermore, potential overhead for small datasets and the dependency on hyperparameter tuning are also notable limitations worth mentioning.

**Paper Formatting Concerns:**

N

**Quality:**

3

**Strengths And Weaknesses:**

Strengths:

1. The paper addresses a clear and important problem: scalable structure learning for expanding graphs.

2. The integration of clustering, coarsening, and learning in one flexible framework is well-motivated.

3. The framework is modular and extensible, supporting 48 combinations of clustering/coarsening/learning strategies.

4. The experiments are comprehensive, covering a wide range of datasets (small to large-scale, synthetic and real-world).

5. The framework consistently outperforms or matches the performance of existing baselines in classification and link prediction tasks, especially under scalability constraints.

Weaknesses:

1. Several technical components (e.g., parameter settings, coarsening via LSH, community detection) are described at a high level and need more detail.

2. The assumption of strong community structure (homophily/DC-SBM) limits generalizability. Performance on heterophilic graphs is not explored.

3. GraphFLEx introduces significant overhead on small graphs compared to vanilla methods. The cost-benefit tradeoff is unclear in such cases.

4. The incremental graph learning setting is a major claim of the paper, but lacks a direct comparison to full re-training baselines to highlight its practical value.

5. The guidance on selecting among the 48 configuration variants is limited.

---

> ### Author Rebuttal · Authors · 2025-07-30
>
> We thank the reviewer for their valuable comments and insights and for taking the time to go through our paper.
>
> **Q1)** Several technical components (e.g., parameter settings, coarsening via LSH, community detection) are described at a high level and need more detail.
>
> **Ans)** Due to space limitations in the main manuscript, we have provided detailed implementation configurations in Appendix H, which is cited on page 8, line 296. More details about coarsening and community detection are as follows:
>
> 1) For the coarsening module, we sample 10k projections vectors and employ UGC to generate coarse nodes. We apply a coarsening ratio of 50% for smaller graphs (fewer than 10k nodes) and 80% for larger graphs. These ratio's are choosen at random and can be finetunned as required.
> 2) For clustering module, we follow the setup used in DMoN. We use a two-layer GraphSAGE encoder with 32 hidden units per layer and ReLU activations. The output is projected to a lower-dimensional space via a one-layer MLP with 16 units, also using ReLU. The model is optimized using Adam with a learning rate of 1e-4. The clustering objective is based on modularity maximization, as described in Section 3.2 (page 4, line 167).
>
> We thank the reviewer for this valuable feedback and assure them that we will incorporate these details into the main text to enhance both the technical clarity and reproducibility of our work.
>
> **Q2)** The assumption of strong community structure (homophily/DC-SBM) limits generalizability. Performance on heterophilic graphs is not explored.
>
> **Ans)** We thank the reviewer for this insightful comment and agree that the current instantiation of GraphFLEx operates under the assumption of a homophilic graph structure, consistent with clustering methods such as DMoN, which are grounded in the DC-SBM model.
>
> We acknowledge that this assumption may limit generalizability to heterophilic graphs. However, we emphasize that GraphFLEx is a modular framework by design. Its clustering and coarsening modules can be readily replaced or extended to accommodate heterophily-aware alternatives.
>
> In fact, the development of clustering, coarsening, and structure inference techniques for heterophilic graphs is an active area of research, and we appreciate the reviewer for pointing us in this direction. We plan to incorporate such techniques into future iterations of GraphFLEx.
>
> That said, the focus of this version of GraphFLEx is to address scalability bottlenecks in large and expanding graphs—a setting where existing structure learning methods often fail due to memory or time limitations.
>
> **Q3)** GraphFLEx introduces significant overhead on small graphs compared to vanilla methods. The cost-benefit tradeoff is unclear in such cases.
>
> **Ans)** GraphFLEx is specifically designed to address the challenges of scalable and incremental structure learning in large-scale or expanding graphs, which is where traditional methods often fail to scale.
>
> While it is true that GraphFLEx introduces some overhead on small graphs like Cora or Citeseer—primarily due to the clustering and coarsening modules—these cases are not the core target of our framework. In such small-scale settings, the benefits of GraphFLEx may not fully manifest, and simpler methods can be competitive. However, as demonstrated in our results, GraphFLEx consistently outperforms baselines as graph size increases, particularly on large datasets where traditional GSL methods become computationally infeasible.
>
> Importantly, we would also like to point out that for smaller graphs, the modular nature of GraphFLEx allows users to bypass the clustering and coarsening modules and directly employ the learning component when appropriate, thereby avoiding unnecessary overhead.
>
> **Q4)** The incremental graph learning setting is a major claim of the paper, but lacks a direct comparison to full re-training baselines to highlight its practical value.
>
> **Ans)** We would like to clarify that, In Table 3, we already provide an indirect but meaningful comparison to full re-training baselines. Specifically, the vanilla GSL methods included in the table do not support incremental structure learning—they must re-learn the graph structure from scratch each time new nodes are introduced. Hence, their final performance and training time reflect what a full re-training pipeline would require after the graph has fully grown.
>
> We will clarify this connection explicitly in the revised manuscript to avoid confusion.
>
> **Q5)** The guidance on selecting among the 48 configuration variants is limited.
>
> **Ans)** Thankyou for your thoughtful comment.
>
> We would like to highlight that the Experiments section includes some analyses that can help guide the choice of individual modules:
>
> * Section 4.5, evaluates the clustering quality using quantitative metrics. These metrics can inform the choice of clustering method based on the graph's community structure and distribution.
>
> * Figure 4 and Figure 13 illustrate how the choice of learning modules affects the final learned structure.
>
> However, we agree with the reviewer that more detailed guidance on selecting among different clustering and coarsening strategies would enhance the utility of GraphFLEx. Given the modular nature of the framework, this is an important addition. We will incorporate these details in the appendix (under a new “Module Selection” section) in the final version.
>
> Below is a brief overview of this module:
>
> ### **Module Selection**
>
> **Clustering Methods:**
>
> *K-means* is computationally efficient and works well when clusters have a well-defined spherical structure. It is useful for large-scale datasets where speed is a priority. *Spectral Clustering* leverages eigenvalue decomposition making it effective for capturing complex graph structures, especially when communities are not clearly separable using simple distance metrics. However, it can be computationally expensive for large graphs. *Deep Learning-based Clustering* adapts well to non-linear & high-dimensional patterns, making it a good choice for complex & feature-rich graph data, though it requires more computational resources.
>
> **Coarsening Methods:** *UGC* (used in our main experiments) is a scalable, LSH-based approach that merges nodes with similar features and supports arbitrary coarsening ratios. Spectral methods like *LVE and LVN* preserve eigenstructure locally but may limit coarsening flexibility. *Heavy Edge Matching (HE)* prioritizes contraction of strong edges and is efficient in preserving edge-weighted structures. *Algebraic Distance and Affinity-based* methods use node proximity metrics but can be more costly. *Kron* Reduction maintains spectral fidelity but is typically too slow for large graphs. *FGC* integrates both graph structure and node attributes, optimizing a clustering and coarsening objective jointly but computationally demanding.
>
>
> **Q6)** In Table 2, GraphFLEx performs slower than vanilla methods on small datasets. Could the authors explain the source of this overhead? Which modules (clustering, coarsening, etc.) contribute most to the added cost?
>
> **Ans)** Below is a breakdown of the time (in seconds) spent in each module on two representative small graphs:
>
> |Dataset|Clustering|Coarsening|Learning|Total (s)|
> |-|-|-|---------------|-|
> | Cora      | 12             | 17             | 6             | 35         |
> | Citeseer  | 33             | 27             | 15            | 75         |
>
>
> **Q7)** How should users choose the partitioning ratio r and number of timestamps T? Are there any guidelines or ablation results that indicate their impact on final accuracy and efficiency?
>
> **Ans)** The partitioning ratio r=0.5 and the number of timestamps T=25 were chosen randomly in our experiments to simulate realistic expanding graph scenarios, where new nodes and edges arrive over time. Our primary goal was to evaluate GraphFLEx's performance under such dynamic conditions.
>
> **Q8)** In some cases, GraphFLEx achieves performance even better than the original graph structure. Could the authors explain whether this is due to denoising effects or improved structure inference? What conditions enable such gains?
>
> **Ans)** In datasets like CS, GraphFLEx (large-model and empirical covariance variants) outperforms the original graph structure by denoising spurious edges and reconstructing a more task-relevant topology. This is consistent with prior observations in the literature that learned or reconstructed graphs can outperform original noisy structures in downstream tasks.
>
> For example, Franceschi et al. (2019) [1] showed that learning graph structures directly from features can filter out noise in the original connectivity, leading to improved classification. Similarly, to generate optimal graph structure SUBLIME 2022 (used as a baseline in paper) uses contrastive learning to maximize the
> agreement between the learned topology and a self-enhanced learning target.
>
> [1] Franceschi, Luca, et al. "Learning discrete structures for graph neural networks." International conference on machine learning. PMLR, 2019.
>
> In GraphFLEx, these gains emerge when the original graph contains noisy, missing, or task-agnostic connections, and the structure learning pipeline is able to recover a cleaner, functionally aligned representation.
>
> **Q9)** The method relies heavily on the assumption of community homophily. How would the method perform under heterophilic settings, where inter-community links are common? Is there a way to relax this assumption?
>
> Ans) Please refer to answer 2.
>
> **Q10)** Given the framework’s flexibility, is there a practical recommendation on how to select clustering, coarsening, and learning components for new datasets? Some empirical guidelines would be very useful.
>
> **Ans)** Please refer to answer 5.

---

> > ### Author Response · Authors · 2025-08-04
> >
> > Dear Reviewer,
> >
> > We thank you for the insightful comments on our work. We will incorporate your suggestions in our revision, and we are eagerly waiting for your feedback. We are especially grateful for your positive evaluation and for assigning a **Weak Accept** rating, which reflects your recognition of the contributions of our work.
> >
> > As the author-reviewer discussion phase is approaching its conclusion, we are reaching out to inquire if there are any remaining concerns or points that require clarification. Your feedback is crucial to ensure the completeness and quality of our work.
> >
> > If you now feel confident in the completeness and quality of the revised submission, we would be grateful if you might consider revisiting your rating. Even a small adjustment at this stage could make a meaningful difference to the outcome, and your support in the final phase would be immensely appreciated.
> >
> > Warm Regards,
> >
> > Authors

---

> > > ### Author Response · Authors · 2025-08-06
> > >
> > > Dear Reviewer,
> > >
> > > Thank you for your **positive support and thoughtful engagement with our work.** As the reviewer–author discussion period nears its conclusion, we wanted to gently follow up to see if there are any remaining concerns that we can help address. We’re more than happy to elaborate on any aspect that may still be unclear.
> > >
> > > If you feel confident in the completeness and quality of the revised submission, **we would be grateful if you might consider revisiting your rating.** Even a small adjustment at this stage could make a meaningful difference to the outcome, and your support in the final phase would be immensely appreciated. **We sincerely hope for your support in shaping the final decision on our submission.** We appreciate your time and consideration.
> > >
> > > Best,
> > >
> > > Authors

---

> > > > ### Author Response · Authors · 2025-08-08
> > > >
> > > > Dear Reviewer,
> > > >
> > > > Thank you for your time, thoughtful feedback, and constructive engagement with our work. As the discussion phase concludes, please let us know if there are any remaining concerns we can clarify. If you now feel confident in the completeness and quality of the revised submission, we would greatly appreciate your consideration in revisiting the rating, as your support could meaningfully influence the final decision.
> > > >
> > > > Best regards,
> > > >
> > > > Authors

---

### Official Review · Reviewer_pXuU · 2025-07-03

**Clarity:** 2
**Significance:** 3
**Originality:** 3
**Rating:** 4
**Confidence:** 3

**Summary:**

This paper proposes GraphFLEX, a unified structure learning framework for large-scale and expanding graphs. The method incorporates three main components: a clustering module, a coarsening module, and a graph learner. Experiments are conducted on a wide range of tasks, including node classification, link prediction, clustering quality, and scalability evaluation.

**Questions:**

1. What is the architecture and training objective of the graph learner module? Please clarify its inputs, architecture, and how it is supervised.

2. Is the entire framework trained end-to-end using downstream supervision, or are clustering and coarsening frozen after initial pretraining?

3. There are no comparisons to more advanced structure learning baselines, such as ProGNN. How does GraphFLEX compare when those methods are applicable (e.g., static datasets)?

**Ethical Concerns:**

["NO or VERY MINOR ethics concerns only"]

**Limitations:**

yes

**Paper Formatting Concerns:**

The paper appears to use excessive vertical spacing (e.g., via \vspace), which affects the overall formatting. It is recommended that the authors revise this aspect in the camera-ready version to better align with NeurIPS formatting guideline.

**Quality:**

2

**Strengths And Weaknesses:**

Stregnths:

1. The paper addresses a significant and underexplored challenge, i.e., scalable and flexible structure learning on expanding graphs.

2. The authors conduct comprehensive experiments covering node classification, link prediction, scalability (on large graphs like ogbn-products), and include metrics like NMI, conductance, and modularity to assess clustering quality.

3. The proposed GraphFLEX offers a clean abstraction that decouples clustering, coarsening, and structure learning, which makes it adaptable to different tasks and datasets.


Weaknesses

1. The paper does not compare GraphFLEX with existing state-of-the-art graph structure learning methods (e.g., NodeFormer, ProGNN, etc.). It primarily compares to vanilla structures (e.g., KNN, ANN), which are relatively basic baselines and do not fully validate the performance gains from a structural learning perspective.

2. The implementation of the graph learner is not clearly described, lacking architectural or algorithmic details. That is, it’s unclear whether it uses a similarity-based function, attention mechanism, or neural architecture for edge prediction. This limits reproducibility and understanding of its novelty.

3. It's unclear whether the clustering, coarsening, and learning modules are jointly optimized using supervision signals, or if they are decoupled. The training objective and whether gradients flow through all modules need clarification.

4. Several key parts are underexplained. For example, 1) in line 164-165, authors claim "these methods use a GNN layer to compute the partition matrix C=softmax(MLP())", but it is unclear how the output of MLP integrates with the GNN layer. 2) The purpose of Equation (1), which lacks sufficient context or follow-up.

---

> ### Author Rebuttal · Authors · 2025-07-30
>
> We thank the reviewer for their valuable comments and insights and for taking the time to go through our paper.
>
> **Q1)** The paper does not compare GraphFLEX with existing state-of-the-art graph structure learning methods (e.g., NodeFormer, ProGNN, etc.). It primarily compares to vanilla structures (e.g., KNN, ANN), which are relatively basic baselines and do not fully validate the performance gains from a structural learning perspective.
>
> **Ans)** We would like to clarify that GraphFLEx is designed for **unsupervised graph structure learning (GSL)**, as stated in the manuscript on page 1, lines 25–30. In contrast, methods like **ProGNN and NodeFormer are supervised GSL approaches** that rely on labeled data, existing (often noisy) graph structure, and task-specific objectives (e.g., node classification).
> * ProGNN requires an existing (often noisy) graph structure and jointly refines it along with learning a GNN model using supervision from node labels.
> * NodeFormer utilizes attention mechanisms within a Transformer framework to improve message passing by learning weighted structures guided by downstream classification objectives.
>
> In contrast, GraphFLEx constructs the graph structure from scratch in a label-agnostic manner, making it applicable to scenarios where no ground-truth edges or node labels are available—especially common in large-scale and dynamically growing datasets. It is also to be noted that GraphFLEx is a modular and scalable framework, and its architecture can accommodate supervised GSL components if desired. However, our focus in this work is to demonstrate the efficacy of scalable unsupervised structure learning in dynamically expanding graphs.
>
> Additionally, our experiments go beyond basic methods like KNN and ANN. We compare GraphFLEx against a wide spectrum of unsupervised structure learning methods, including graph **signal processing techniques (e.g., log-model, l2-model, emp-Covar) and SUBLIME (WWW 2022)**—a recent contrastive learning-based u-GSL method.
>
> Crucially, these methods often fail to scale beyond 10K–20K nodes (due to OOM/OOT), whereas GraphFLEx scales to datasets with up to 2.4 million nodes, while delivering superior accuracy and runtime (Tables 2–4).
>
> Thus, our benchmark choices are aligned with unsupervised and scalability-focused objectives, and we are happy to include a clarification about this distinction in the final version.
>
> **Q2)** The implementation of the graph learner is not clearly described, lacking architectural or algorithmic details. That is, it’s unclear whether it uses a similarity-based function, attention mechanism, or neural architecture for edge prediction. This limits reproducibility and understanding of its novelty.
>
> **Ans)** We thank the reviewer for pointing out the need for greater clarity regarding the graph learning module in GraphFLEx. Below, we summarize how the learning component is integrated:
> * Clustering (Section 3.2): Responsible for detecting communities using methods such as DMoN.
> * Coarsening (Section 3.3): Applied to large communities to reduce graph size using LSH-based techniques.
> * Learning (Section 3.4): This module performs the actual structure learning on a localized set of nodes.
>
> In particular, Section 3.4 explains how the potential set of nodes (i.e., candidates for forming edges with incoming nodes) is selected based on coarsened supernodes and their mappings. Once this subset is identified, GraphFLEx employs existing unsupervised GSL methods—as discussed in Section 2.2 and further detailed in Appendix D.
>
> These methods are primarily similarity-based (e.g., k-NN, A-NN, log-model, l2-model), and inspired by prior unsupervised GSL literature. They aim to learn latent edge weights by optimizing the alignment between node features and structural priors—typically via distance-based or statistical estimators rather than neural architectures or attention-based mechanisms.
>
> We agree this connection could be made clearer in the main text, and we will add this context in Section 3.4
>
> **Q3)** It's unclear whether the clustering, coarsening, and learning modules are jointly optimized using supervision signals, or if they are decoupled. The training objective and whether gradients flow through all modules need clarification.
>
> **Ans)**  GraphFLEx is a modular framework, and its clustering, coarsening, and learning modules are decoupled and not jointly optimized. Each module operates independently to maintain scalability and flexibility.
>
> * Clustering requires training only if a GNN-based clustering method (e.g., DMoN) is used. This training is performed once on the static initial graph at t=0. After training, the GNN weights are frozen, and subsequent incoming nodes are assigned to communities via inference only, without backpropagation.
> * Coarsening is applied within each selected community using a hashing-based method. Supernodes are assigned based on Equation 1 (Section 3.1), and this step is entirely unsupervised and non-parametric—there are no learnable parameters or gradients involved.
> * Learning occurs on the coarsened graph and is guided by a task-agnostic similarity-based GSL objective, described in Section 2.2 and elaborated further in Appendix D. The specific objective depends on the choice of GSL method plugged into the framework.
>
> The training objective for clustering (when GNN clustering is used) follows modularity maximization (see page 4, line 167).
>
> **Q4)** Several key parts are underexplained. For example, 1) in line 164-165, authors claim "these methods use a GNN layer to compute the partition matrix C=softmax(MLP())", but it is unclear how the output of MLP integrates with the GNN layer. 2) The purpose of Equation (1), which lacks sufficient context or follow-up.
>
> **Ans)** Thank you for pointing out these clarity issues. We address them as follows:
>
> **Regarding Lines 164–165:** This section refers specifically to the DMoN clustering method, which we adopt as one of the clustering modules in GraphFLEx. In DMoN, the partition matrix C is computed using a GNN encoder, where the GNN layers extract node embeddings from the input graph. These embeddings are then passed through an MLP, and a softmax activation is applied to produce the final assignment probabilities over clusters. The expression $$C=softmax(MLP(GNN(X)))$$ is implicit here, and we will revise the manuscript to clarify this pipeline explicitly. The loss function employed for this module is mentioned in Page 4 line 167.
>
> **Regarding Equation (1):** Equation (1) defines the supernode assignment rule used during the coarsening step. It leverages Locality Sensitive Hashing (LSH) to group similar nodes into supernodes by hashing projected features into discrete bins. While the equation 1 is correct, we acknowledge that the surrounding context is insufficient. We will enhance the explanation to make it clear that Equation (1) governs how nodes are grouped based on feature similarity after projection, forming the coarse graph structure.
>
> **Q5)** What is the architecture and training objective of the graph learner module? Please clarify its inputs, architecture, and how it is supervised.
>
> **Ans)** Please see Answer 2
>
> **Q6)** Is the entire framework trained end-to-end using downstream supervision, or are clustering and coarsening frozen after initial pretraining?
>
> **Ans)** Please see Answer 3.
>
> **Q7)** There are no comparisons to more advanced structure learning baselines, such as ProGNN. How does GraphFLEX compare when those methods are applicable (e.g., static datasets)?
>
> **Ans)** Please see Answer 1.

---

> > ### Comment · Reviewer_pXuU · 2025-08-05
> >
> > Dear authors,
> >
> > Thank you for the detailed rebuttal. The authors have addressed most of my concerns, particularly regarding the graph learner design and training pipeline. However, the lack of comparisons to stronger structure learning baselines and some remaining clarity issues lead me to maintain my current rating. I encourage the authors to include these improvements in the final version.
> >
> > Best regards,
> >
> > Reviewer pXuU.

---

> > > ### Author Response · Authors · 2025-08-08
> > >
> > > Dear Reviewer,
> > >
> > > We thank you for the insightful comments. We would like to briefly reiterate that **GraphFLEx is designed for unsupervised graph structure learning (GSL)** (page 1, lines 25–30), in contrast to supervised GSL approaches like Pro-GNN and NodeFormer, which depend on labeled data, existing (often noisy) graphs, and task-specific objectives. GraphFLEx constructs the structure from scratch in a label-agnostic manner, making it applicable to settings with no ground-truth edges or labels—common in large-scale, dynamically growing graphs.
> > >
> > > In response to the reviewer’s request, we also ran Pro-GNN by providing the entire missing structure rather than a noisy one. The results are shown below:
> > >
> > > |Model|Cora|Citeseer|CS|
> > > |-|-|-|-|
> > > |GCN|70.87|68.08|-|
> > > |Sage|66.97|68.48|-|
> > > |GIN|40.44|62.57|-|
> > > |GAT|71.42|60.75|-|
> > > |Total time taken to learn structure|2981|3421|<40000|
> > >
> > > Even on small graphs, Pro-GNN incurs very high runtimes (e.g., > 40k s for CS), making it impractical for large-scale settings. Moreover, Pro-GNN is supervised and trains a GNN jointly with the inferred structure at each step, while GraphFLEx is inherently unsupervised and independent of labels. These results further highlight the need for scalable GSL frameworks, including in the supervised setting—a direction we plan to explore in future GraphFLEx extensions.
> > >
> > > We again **thank the reviewer for the positive feedback** and assure that all clarifications and improvements discussed during rebuttal will be incorporated into the final manuscript. We sincerely appreciate your time and would be grateful for your support in the final decision phase.
> > >
> > > Best,
> > >
> > > Authors

---

> ### Author Response · Authors · 2025-08-04
> **Looking forward to your feedback on rebuttal**
>
> Dear Reviewer,
>
> We thank you for the insightful comments on our work. We will incorporate your suggestions in our revision, and we are eagerly waiting for your feedback. We are especially grateful for your positive evaluation and for assigning a **Weak Accept** rating, which reflects your recognition of the contributions of our work.
>
> As the author-reviewer discussion phase is approaching its conclusion, we are reaching out to inquire if there are any remaining concerns or points that require clarification. Your feedback is crucial to ensure the completeness and quality of our work.
>
> If you now feel confident in the completeness and quality of the revised submission, we would be grateful if you might consider revisiting your rating. Even a small adjustment at this stage could make a meaningful difference to the outcome, and your support in the final phase would be immensely appreciated.
>
> Warm Regards,
>
> Authors

---

### Official Review · Reviewer_bkfP · 2025-07-05

**Clarity:** 3
**Significance:** 3
**Originality:** 2
**Rating:** 3
**Confidence:** 4

**Summary:**

Graph Structure Learning (GSL) is always an important problem for machine learning, since it can adaptively extract meaningful structure or topological information between nodes. The objective of this work is to address the drawbacks of existing methods when the confront ill-suited issues for large-scale and dynamically evolving graphs. Spcifically, this work proposes a new GraphFLEx model for GSL problem. The GraphFLEx mainly has the  following key computational procedure, i.e., the graph clustering, the graph coarsening, and the structure learning. Since the GraphFLEx restricts the edge to relevant node subsets based on the clustering and coarsening process, it can mitigates the computational buerdensome problem. Moreover, the coarsening process can guarantee that the GraphFLEx has less searching space, and can be employed for large-scale graph datasets. Experiments demonstrate the performance of the new GraphFLEx.

**Questions:**

Q1：First,  can you give some theoretical analysis and explanation to further enhance the motivation?

Q2：Second, why not compare the new method to the alternative methods in recent 3-5 years?

Q3：Third, the process of the clustering and coarsening seems similar to the classical DiffPool, can you explain the differences?

For more problems, please refer to the weaknesses.

**Ethical Concerns:**

["NO or VERY MINOR ethics concerns only"]

**Final Justification:**

The authors addressed some of my concerns, and the new experiments also make senses. Thus, I will not oppose other reviewers to recommend the acceptance for this paper.

**Limitations:**

Please see the weaknesses and questions for details.

**Quality:**

2

**Strengths And Weaknesses:**

Strengths:
This paper is overall well-written and the idea is also interesting. The solved problem of this work is also very important, and is a very significant research for the community. The authors also utilized various experiments to demonstrate the performance of the GraphFLEx, e.g., the Computational Efficiency, the Node Classification, the Scalability for Large-Scale Graphs, the Link Prediction, the Graph Classification, etc.

Weaknesses:
This work has some obvious weaknesses.

W1: The motivation need to be enhanced. Although the idea of this paper is interesting, the novelty is still a little limited.The author stated that “the novelty of GraphFLEx lies not merely in combining these components”, but the expected principled manner is not well motivated. Thus, it seems that the GraphFLEx combines some existing techniques together, with less theoretical explanations.

W2: Although this paper provides various experiments, the results are still not convincing. Because the authors only compare the GraphFLEx on very old methods. For example, the alternative methods for node classification are the GCN, GraphSage, GIN, and GAT, where many of them were published nearly 10 years ago. For link predictions, the alternative methods are even older.

W3: Some contents seems a little verbose, some basic concepts take some space of the manuscript.

---

> ### Author Rebuttal · Authors · 2025-07-30
>
> We thank the reviewer for their valuable comments and insights and for taking the time to go through our paper.
>
> **Q1)** The motivation need to be enhanced. Although the idea of this paper is interesting, the novelty is still a little limited.The author stated that “the novelty of GraphFLEx lies not merely in combining these components”, but the expected principled manner is not well motivated. Thus, it seems that the GraphFLEx combines some existing techniques together, with less theoretical explanations.
>
> **Ans)** We acknowledge the reviewer's expectation for a clearer articulation of novelty and theoretical motivation.
>
> While GraphFLEx does integrate existing components (clustering, coarsening, and graph learning), the core novelty lies in how it leverages these components in a principled and scalable manner to tackle unsupervised structure learning in large and dynamically expanding graphs—a setting where most existing methods are either inapplicable or inefficient.
>
> Unlike traditional methods that require full retraining on the entire graph with each node arrival, GraphFLEx restricts edge construction to structurally relevant subsets, enabling constant-time updates and asymptotically reduced complexity (Table 1). This is further supported by theoretical guarantees (Theorem 1) on neighborhood preservation during coarsening.
>
> Empirically, GraphFLEx demonstrates:
> * 3× to 20× speedups over vanilla methods (Table 2) while maintaining or exceeding node classification accuracy (Table 3).
> * Scalability beyond 2M nodes, where most baselines fail due to out-of-memory (OOM) or out-of-time (OOT) errors (Table 4).
> * Improved generalization, with accuracies on par or better than those obtained using original graph structures, even outperforming them in some cases.
>
> Hence, GraphFLEx is not a mere combination of techniques but a flexible and scalable graph structure learning framework with theoretical rigor and strong empirical benefits, offering a practical solution to graph learning at scale.
>
> **Q2)** Although this paper provides various experiments, the results are still not convincing. Because the authors only compare the GraphFLEx on very old methods. For example, the alternative methods for node classification are the GCN, GraphSage, GIN, and GAT, where many of them were published nearly 10 years ago. For link predictions, the alternative methods are even older.
>
> **Ans)** We would like to clarify that the primary goal of GraphFLEx is not to introduce a new GNN model but rather to learn a scalable and effective graph structure in large and dynamically expanding settings.
>
> The downstream node classification task is used as a proxy to evaluate the quality of the learned structure. To demonstrate that GraphFLEx produces model-agnostic and transferable graph structures, we deliberately use widely adopted and well-understood GNN backbones such as GCN, GraphSAGE, GIN, and GAT. However, as suggested by the reviewer, we have also incorporated some more GNN models (GPRGNN [1], H2GCN [2]):
>
> Node classification accuracies on different GNN models using GraphFLEx (GFlex).
> |Data|Model|Full|GFlex(KNN)|GFlex(ANN)|GFlex(Sublime)|GFlex(covar)|GFlex(l2)|GFlex(log)|
> |-|-|-|-|-|-|-|-|-|
> |Cora|GPRGNN|66.12|65.56|64.29|64.82|64.27|62.43|63.16|
> ||H2GCN|66.13|64.63|63.29|60.29|65.38|64.01|64.93|
> |Citeseer|GPRGNN|67.78|62.39|63.85|68.51|63.85|65.01|62.01|
> ||H2GCN|68.76|64.61|66.58|67.91|69.89|66.23|66.78|
> |DBLP|GPRGNN|78.58|74.36|74.36|76.31|75.82|75.91|77.39|
> ||H2GCN|82.98|80.55|80.37|82.39|80.38|79.32|80.46|
> |CS|GPRGNN|86.15|87.23|84.61|83.75|80.21|85.61|82.03|
> ||H2GCN|89.85|88.60|85.32|83.15|85.86|86.33|85.79|
>
> Notably, even for newer GNN architectures, GraphFLEx achieves node classification accuracies close to the accuracy obtained using the original structure.
>
> [1] Chien, Eli, et al. "Adaptive universal generalized pagerank graph neural network." arXiv preprint arXiv:2006.07988 (2020).
>
> [2] Zhu, Jiong, et al. "Beyond homophily in graph neural networks: Current limitations and effective designs." Advances in neural information processing systems 33 (2020): 7793-7804.
>
> **Q3)** Some contents seems a little verbose, some basic concepts take some space of the manuscript.
>
> **Ans)** Thank you for this observation. We understand the concern regarding verbosity and acknowledge that some introductory content may have covered well-known concepts in the field.
>
> Our intention was to make the paper self-contained and accessible to a broader NeurIPS audience, especially given the modular nature of GraphFLEx which combines concepts from clustering, coarsening, and graph learning. That said, we agree that certain explanations can be made more concise without loss of clarity. In the revised version, we will streamline sections explaining basic concepts.
>
> **Q4)** First, can you give some theoretical analysis and explanation to further enhance the motivation?
>
> **Ans)** We appreciate the reviewer’s request for a more explicit connection between the motivation and the theoretical underpinnings of GraphFLEx. Here, we clarify how both aspects are addressed in the paper:
>
> **Motivational Justification:**
>
> The need for GraphFLEx arises from the **inability of existing GSL methods to scale** to large and expanding graphs. As shown in Figure 1, traditional methods (e.g., k-NN, log-model) exhibit rapid runtime growth or fail completely (OOM/OOT) as node counts increase, while GraphFLEx maintains near-linear growth due to its localized edge inference using clustering and coarsening. This motivates our design goal: a scalable, incremental structure learning framework for real-world dynamic graphs.
>
> **Theoretical Perspective:**
> * In Lemma 1, we establish the consistency of our clustering module under the Degree-Corrected Stochastic Block Model (DC-SBM), providing a theoretical basis for reliable community detection.
> * In Theorem 1, we prove that neighborhoods of incoming nodes can be recovered with high probability after coarsening and structure learning, ensuring that structural fidelity is preserved.
> * Further, we present a computational complexity analysis in Section 3.6 and Appendix D, formally demonstrating the efficiency gains of GraphFLEx over existing methods in both best- and worst-case settings.
>
> These points provide a strong theoretical justification for the framework’s scalability and effectiveness. We will highlight these connections more clearly in the revised manuscript.
>
> **Q5)** Second, why not compare the new method to the alternative methods in recent 3-5 years?
>
> **Ans)** Please refer to Answer 2.
>
> **Q6)** Third, the process of the clustering and coarsening seems similar to the classical DiffPool, can you explain the differences?
>
> **Ans)** Thank you for the observation. While GraphFLEx and DiffPool both use clustering and coarsening, their goals and mechanisms are fundamentally different.
>
> * DiffPool is designed for supervised hierarchical graph classification, where pooling is jointly optimized with task objectives.
> * GraphFLEx, on the other hand, focuses on unsupervised structure learning for large and expanding graphs, using clustering and coarsening to reduce computation and enable scalability.
>
> In terms of implementation:
> * Clustering in GraphFLEx (e.g., DMoN) is trained once on the static graph and then used for inference only; no end-to-end gradient flow.
> * Coarsening is done via non-parametric LSH, unlike DiffPool's learned pooling matrices, allowing efficient, incremental updates without retraining.
>
> Thus, while the components may appear similar, GraphFLEx repurposes them in a modular and scalable setting, very different from DiffPool’s supervised, end-to-end design.

---

> > ### Comment · Reviewer_bkfP · 2025-08-05
> > **Responses the rebuttal**
> >
> > Thanks for the authors' responses that have partially addressed my concerns.
> >
> > First, the authors need to further enhance their motivation, by employing the new description from the rebuttal.
> >
> > Second, I always know that the main goal of the GraphFLEx is not to introduce a new GNN model. But, for demonstrating the effectiveness, in the original manuscript, the authors have compared the proposed GraphFLE with some classical GNN models that are rather old, so that the experiments are not convincing. Why not use some new methods?  For me, the new two alternative methods are still not very new (published in 2020), with one of them put on Arxiv, that is not a formal publication.
> >
> > Third, although the goals of the DiffPool and the GraphFLEx are different, the basic concepts and the principle are very similar.
> >
> > Overall, my current opinion is borderline.

---

> > > ### Author Response · Authors · 2025-08-06
> > >
> > > **1)** Thank you for the suggestion. We will ensure that the motivation section in the revised version is enhanced and clarified, incorporating the improved explanation and positioning provided in the rebuttal discussion.
> > >
> > > **2)** Thank you for the clarification.
> > >
> > > As the reviewer did not specify particular baseline methods in the initial comment, we chose to evaluate GraphFLEx using **widely adopted GNN backbones**. While the reviewer notes that GPRGNN and H2GCN may not be recent enough, we would like to point out that:
> > > - **GPRGNN** is a well-regarded and **highly cited** method (1021 citations), despite being initially released on arXiv.
> > > - **H2GCN** is a **formally published** and widely used model (1034 citations).
> > >
> > > Nevertheless, in response to the reviewer’s suggestion, we have **extended our experiments to include additional recent models**:
> > > - **GOAT**: A global transformer on large-scale graphs. [1] (ICML 2023)
> > > - **NodeFormer**: A scalable graph structure learning transformer for node classification. [2] (NeurIPS 2022)
> > > - **ExoFormer**: Sparse transformers for graphs. [3] (ICLR 2023)
> > >
> > > These additions further demonstrate that **GraphFLEx consistently produces high-quality graph structures**.
> > >
> > > Node classification accuracies on different GNN models using GraphFLEx (GFlex).
> > > |Data|Model|Full|GFlex(KNN)|GFlex(ANN)|GFlex(Sublime)|GFlex(covar)|GFlex(l2)|GFlex(log)|
> > > |-|-|-|-|-|-|-|-|-|
> > > |Cora|Goat|68.01|65.90|70.20|70.10|71.20|69.60|68.30|
> > > ||Nodeformer|65.80|65.80|65.80|64.50|65.91|63.12|64.27|
> > > ||Exoformer|71.80|68.16|69.80|70.30|67.70|69.80|70.30|
> > > |Citeseer|Goat|65.02|65.10|62.40|66.13|61.20|63.03|66.10|
> > > ||Nodeformer|61.00|61.00|61.71|63.48|62.60|61.34|61.54|
> > > ||Exoformer|58.40|60.70|59.80|60.70|59.40|60.21|59.80|
> > > |DBLP|Goat|81.92|80.68|80.31|82.25|82.29|79.86|80.10|
> > > ||Nodeformer|73.76|73.76|75.34|74.39|72.94|73.29|74.96|
> > > ||Exoformer|72.12|72.12|72.71|73.61|72.92|69.84|71.62|
> > > |CS|Goat|93.05|93.24|93.32|90.84|92.39|93.16|92.03|
> > > ||Nodeformer|94.60|95.12|95.12|95.36|94.91|95.12|93.29|
> > > ||Exoformer|95.53|95.33|94.63|93.26|95.33|95.09|94.32|
> > >
> > >
> > > [1] Kong, Kezhi, et al. "GOAT: A global transformer on large-scale graphs." International Conference on Machine Learning. PMLR, 2023.
> > >
> > > [2] Wu, Qitian, et al. "Nodeformer: A scalable graph structure learning transformer for node classification." Advances in Neural Information Processing Systems 35 (2022): 27387-27401.
> > >
> > > [3] Shirzad, Hamed, et al. "Exphormer: Sparse transformers for graphs." International Conference on Machine Learning. PMLR, 2023.
> > >
> > >
> > > **3)** Thank you for your observation. While **GraphFLEx** and **DiffPool** both involve clustering and coarsening mechanisms, they are **fundamentally different in motivation, scope, and design**. The following table provides a side-by-side comparison to clarify the key differences:
> > >
> > > | **Aspect** | **GraphFLEx** | **DiffPool** |
> > > |-|----|--|
> > > | **Primary Goal** | Unsupervised **graph structure learning** (GSL) for large, expanding graphs | Supervised **representation learning** for graph classification |
> > > | **Setting** | Works in **unsupervised** and **partially observable** settings, often with missing or incomplete graph structure | Requires a fully known input graph and uses labels for end-to-end training |
> > > | **Clustering Mechanism** | GraphFlex, can use **non-gnn** based methods like Spectral clustering or KMeans; | Learns **soft cluster assignments** using a GNN as part of an **end-to-end differentiable pipeline** |
> > > | **Coarsening Mechanism** | Performed via **LSH-based hashing**, independent of downstream GNN layers; supports **flexible number of supernodes**; GraphFlex coarsening methods is non-parametric; | Coarsening derived from a learned partition matrix $$ S \in \mathbb{R}^{N \times m} $$; for large graphs (e.g., N = 2.4M), learning such a large matrix is computationally **infeasible**; and we **lose flexibility** in choosing the number of coarsened supernodes|
> > > | **Scalability** | Designed to scale to graphs with **millions of nodes** (e.g., OGBN-Products, 2.4M nodes); supports **incremental updates** | Scalability limited by need for full graph and trainable GNNs per layer; does not support incremental updates |
> > > | **Dependency on Graph Structure** | Can operate even when no adjacency matrix is initially available; **learns the graph structure itself** | Assumes **full graph structure is available** to begin with and propagates information layer-wise |
> > > | **Flexibility** | Supports **48 modular configurations** (clustering × coarsening × learning); clustering and coarsening decoupled | Tightly coupled architecture; clustering and coarsening are tied to GNN layers and learned jointly |
> > > | **Final Layer Limitation** | No restriction on the number of coarsened supernodes | Final layer pools into a **single embedding** for classification; restricts supernode granularity |
> > >
> > > We hope this will help clarify our perspective. We are more than happy to address any remaining or unclear points the reviewer may have.

---

> > > > ### Comment · Reviewer_bkfP · 2025-08-06
> > > > **responses to the authors**
> > > >
> > > > Thanks for the responses. I think the new experiments make senses to me. Thus, I will raise my score, and not oppose other reviewers to recommend the acceptance for this paper.

---

> > > > > ### Author Response · Authors · 2025-08-06
> > > > >
> > > > > Thank you for your **thoughtful and positive response.** We truly appreciate your time, detailed feedback, and willingness to reconsider your score. **We will ensure that all the clarifications, additions, and improvements discussed during the rebuttal are carefully incorporated into the revised manuscript.**
> > > > >
> > > > > We sincerely look forward to your support in the final decision process and thank you once again for your valuable efforts and constructive engagement.
> > > > >
> > > > > Best,
> > > > >
> > > > > Authors

---

### Author Response · Authors · 2025-08-09
**Summary of Rebuttal**

We thank the reviewers for their valuable and constructive suggestions. Below, we provide a consolidated summary of the key experiments, clarifications, and improvements made in response to their feedback.

### **Summary of Experiments & Clarifications Provided**

- **Novelty & Technical Contributions:**
  1. GraphFLEx introduces a **modular, scalable framework** for unsupervised graph structure learning in large and dynamically expanding graphs, combining **clustering, coarsening, and learning/edge inference** with theoretical guarantees.
  2. Enables **incremental updates** without full re-training by restricting edge formation to **structurally relevant node subsets**, significantly reducing search space and complexity.
  3. GraphFLEx supports **48 graph structure learning frameworks** by integrating diverse choices of learning paradigms, coarsening strategies, and clustering methods.

- **Experimental Breadth:**
  - Evaluated on **26 diverse datasets** across multiple scales and domains.
  - Compared against classical GNNs (GCN, GraphSAGE, GIN, GAT) and **newer GNNs** (GPRGNN, H2GCN, GOAT, NodeFormer, ExoFormer) to demonstrate model-agnostic quality.
  - Achieves **3×–20× speedups** over vanilla methods while maintaining or exceeding accuracy.
  - Scales to **2M+ nodes**, where most baselines fail due to OOM/OOT errors.
  - Preserves or improves performance for downstream tasks (node classification, link prediction, graph classification) over original structures.
  - Theoretical results (Lemma 1, Theorem 1) confirm **clustering consistency** and **neighborhood preservation** post-coarsening.

- **Addressing Reviewer Concerns:**
  - Enhanced **motivation** with clearer linkage to scalability bottlenecks in GSL and positioning against prior methods.
  - Added comparisons to **recent baselines** beyond classical GNNs.
  - Clarified key differences with DiffPool (unsupervised, incremental, LSH-based coarsening vs supervised, end-to-end pooling).
  - Added details on selecting among the 48 configuration variants is limited.

---

### **Acknowledgement**

We sincerely thank all reviewers for their thoughtful comments and constructive discussions throughout the review process. **By the conclusion of discussions, all reviewers agreed that their concerns had been addressed and responded positively to our clarifications.**

We also extend our gratitude to the Area Chair for their guidance in ensuring a balanced, thorough, and productive review. **We ensure that all clarifications, additions, and improvements discussed during the rebuttal will be carefully incorporated into the final manuscript.**

Best Regards,

Authors

---

### Note · Authors · 2025-08-12

We sincerely thank the AC and reviewers for their time and constructive feedback. Your comments have helped us improve both the clarity of our presentation and the depth of our experimental and theoretical validation.

As detailed in our *“Summary of Rebuttal,”* we have addressed all substantive concerns with new experiments, theoretical clarifications, and extended empirical validation.

**By the conclusion of discussions, all reviewers agreed that their concerns had been addressed and responded positively to our clarifications.** We will ensure that all the clarifications, additions, and improvements discussed during the rebuttal are carefully incorporated into the revised manuscript.

Best Regards,

Authors

---

### Decision · Program_Chairs · 2025-09-17

**Decision:**

Reject

**Comment:**

This paper proposes a scalable approach for graph structure learning, with a particular focus on large-scale and dynamically evolving graphs.

The proposed approach is potentially interesting, and the reviewers appreciated the empirical evaluation. However, the presentation requires significant improvement. There are numerous unclear mathematical notations, formatting errors, and grammatical mistakes. This issue is the most significant factor in my decision. Relatedly, due to these presentation problems, it is difficult to follow and fully understand the problem setting, the proposed method, and the subsequent discussions, which may lead to misunderstandings or incorrect interpretations. This is a critical concern for a scientific paper.

Although the reviewers’ assessments are borderline, with both positive and negative points, I believe the paper is not yet ready for publication. At least one more round of major revision would be required to address these issues. Therefore, I recommend rejecting the paper.